# CHAIN-OF-GOALS HIERARCHICAL POLICY FOR LONG-HORIZON OFFLINE GOAL-CONDITIONED RL

## ABSTRACT

Offline goal-conditioned reinforcement learning remains challenging for long-horizon tasks. While hierarchical approaches attempt to address this by decomposing tasks into high-level subgoals, most existing methods rely on two-level architectures with separate networks, leading to fundamental limitations: they generate only a single intermediate subgoal, leading the low-level policy to act without awareness of the final goal when misled by erroneous subgoals, and prevent end-to-end optimization due to separate training objectives. We discover a novel solution to these challenges through chain-of-thought reasoning from large language models. Building on this insight, we introduce the Chain-of-Goals Hierarchical Policy (CoGHP), a new framework that reformulates hierarchical control as autoregressive sequence generation within a single unified architecture. CoGHP generates a sequence of latent subgoals and the primitive action in a single forward pass, where each subgoal acts as a "reasoning token" encoding intermediate decision-making. To implement this chain-of-thought approach in hierarchical RL, we pioneer the use of the MLP-Mixer architecture. This design enables efficient cross-token communication through simple feedforward operations and captures consistent structural relationships essential for hierarchical reasoning. Experimental results on challenging navigation and manipulation benchmarks demonstrate that CoGHP consistently outperforms strong baselines, demonstrating its effectiveness for long-horizon offline control tasks.

## 1 INTRODUCTION

Offline goal-conditioned reinforcement learning (RL) (Chebotar et al. (2021); Yang et al. (2022); Ma et al. (2022)) aims to learn a policy that reaches specified goals using only static, pre-collected datasets, which is useful when interactions with environments are costly or unsafe. However, as horizons expand, the gap between optimal and suboptimal action values diminishes due to discounting and compounded Bellman errors, leading to unreliable policy (Park et al. (2023)). Offline hierarchical RL (Gupta et al. (2019); Park et al. (2023); Schmidt et al. (2024)) addresses this by decomposing tasks into high-level subgoal selection and low-level control, but traditional approaches face a fundamental structural limitation. Most existing hierarchical RL methods rely on two-level hierarchical structures with separate networks for high-level and low-level policies. This architectural separation leads to three critical limitations. First, these approaches typically generate only a single intermediate subgoal, making them inadequate for complex tasks that require coordinating multiple intermediate decisions. Second, when the high-level policy generates erroneous subgoals, the low-level policy blindly executes toward these misguided targets. As a result, it loses awareness of the final goal and may select sub-optimal actions. Third, training hierarchy levels under separate objectives prevents end-to-end gradient flow, blocking corrective signals from propagating across decision-making stages and hindering the coordinated multi-stage reasoning necessary for long-horizon tasks.

How can we develop a unified approach that naturally scales to multiple hierarchy levels while maintaining both computational efficiency and learning stability? Rather than adding more separate networks to handle longer horizons, we need a fundamentally different architectural paradigm that can handle multi-step sequences of intermediate decisions within a single, cohesive framework. We discover that the answer to this question lies in the chain-of-thought reasoning paradigm of large language models (LLMs) (Wei et al. (2022); Zhang et al. (2022)), where LLMs break down complex

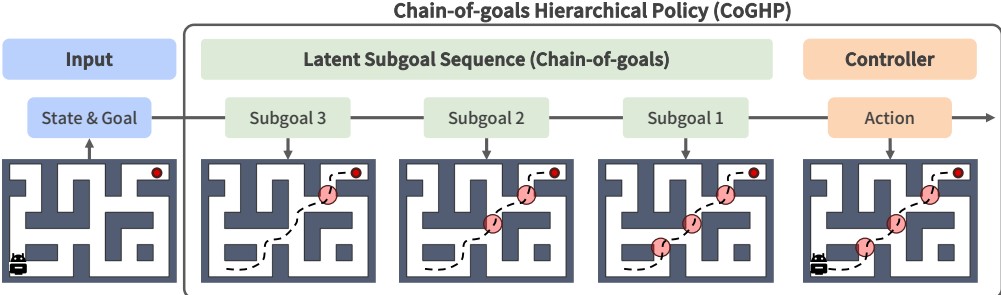

Figure 1: **Chain-of-Goals Hierarchical Policy (CoGHP).** CoGHP autoregressively generates a sequence of latent subgoals and the primitive action within a single forward pass. Each subgoal serves as a reasoning token, providing the agent with sufficient guidance to reach the goal. Autoregressive generation ensures that later predictions build upon earlier ones while maintaining awareness of the final goal. To ensure that the subgoal closest to the agent carries the most informative signal, the sequence is generated in reverse order, beginning with the farthest subgoal relative to the current state and progressing toward the nearest one.

problems by generating sequential intermediate reasoning steps before producing final answers. This paradigm directly addresses the three critical limitations of traditional offline hierarchical RL. LLMs generate multiple intermediate reasoning steps to handle complex multi-step problems (solving the single subgoal limitation), maintain access to the original query throughout the entire reasoning chain (preserve awareness of the final goal), and handle the entire reasoning process within a single unified sequence model (enabling information and training-signal flow across reasoning steps), which are exactly the properties needed for effective hierarchical control in long-horizon RL tasks.

Building on this insight, we introduce the **Chain-of-Goals Hierarchical Policy (CoGHP)**, which incorporates the chain-of-thought paradigm into offline goal-conditioned RL through a novel architectural design. Instead of relying on separate networks for different hierarchy levels, CoGHP reformulates hierarchical control as the autoregressive sequential generation of latent subgoals and the primitive action within a single forward pass (Figure 1). Each latent subgoal functions as a "reasoning token" that encodes an intermediate decision-making step. Autoregressive generation ensures that later predictions build upon earlier ones while preserving access to the final goal. This chain-of-thought-style input–intermediate reasoning steps–primitive action structure has recently emerged as a useful paradigm for robotic control in vision-language-action models (Mu et al. (2023); Zhao et al. (2025)). CoGHP takes a step further by extending this perspective to the offline goal-conditioned RL setting and instantiating it as a unified hierarchical control framework. To effectively realize this sequence modeling, we pioneer the application of the MLP-Mixer architecture to hierarchical RL. Its simple feedforward design enables efficient cross-token communication, making it well-suited for processing a sequence of state, goal, latent subgoals, and action. Finally, we train this unified architecture with a shared value function learned from offline data, which provides training signals for all sequence elements, including both intermediate subgoals and primitive actions. This training strategy allows gradient-based error correction to propagate seamlessly across the entire hierarchy.

In summary, our contributions are threefold. First, we introduce the first framework to bring the chain-of-thought reasoning paradigm from LLMs to offline hierarchical RL, reformulating hierarchical control as autoregressive sequence generation of intermediate subgoals that act as reasoning tokens. Second, we pioneer the application of MLP-Mixer architecture in hierarchical RL, leveraging its token-mixing capabilities and sequential processing strengths to enable unified end-to-end training across all decision-making stages. Third, we demonstrate that CoGHP outperforms strong baselines on challenging navigation and manipulation benchmarks, demonstrating its effectiveness for long-horizon offline control tasks.

## 2 RELATED WORK

**Offline Hierarchical RL**  Prior work in offline RL has primarily tackled distribution shift and overestimation through regularization and constraint-based methods (Kostrikov et al. (2021); Kumar et al. (2020); Wu et al. (2019)), but these approaches still struggle on long-horizon tasks. To

address this issue, offline hierarchical RL decomposes decision-making into temporally abstract subgoals and low-level control. Key directions include skill and primitive discovery from static data (Ajay et al. (2020); Krishnan et al. (2017); Pertsch et al. (2021); Choi & Seo (2025); Pertsch et al. (2022)), latent plan representation learning for efficient high-dimensional planning (Jiang et al. (2022); Rosete-Beas et al. (2023); Lynch et al. (2020); Shah et al. (2021)), integrated hierarchical planners combining subgoal selection with goal-conditioned controllers (Park et al. (2023); Gupta et al. (2019); Schmidt et al. (2024); Li et al. (2022)), and model-based world modeling for offline planning (Shi et al. (2022); Freed et al. (2023)). While these modular pipelines enhance temporal abstraction, they suffer from fundamental architectural limitations, including single subgoal constraints, loss of final goal awareness when misled by erroneous subgoals, and fragmented optimization. In contrast, CoGHP reformulates hierarchical control as unified autoregressive sequence generation, producing multiple subgoals within a single architecture that enables end-to-end optimization.

**Chain-of-Thought** Chain-of-thought prompting was first shown to unlock complex reasoning in large language models by eliciting intermediate rationales, yielding dramatic gains on arithmetic, commonsense, and symbolic benchmarks (Wei et al. (2022)); subsequent work refined its application through analysis of prompting factors and enhanced reasoning via automatic rationale synthesis, self-consistency decoding, and progressive problem decomposition (Sprague et al. (2024); Zhang et al. (2022); Wang et al. (2022a); Zhou et al. (2022)). In robotics and embodied AI, chain-of-thought-inspired intermediate planning has been applied to vision–language agents, navigation, policy learning with semantic subgoals, sensorimotor grounding, and affordance-based action planning (Mu et al. (2023); Lin et al. (2025); Chen et al. (2024); Zawalski et al. (2024); Brohan et al. (2023)). Building on this foundation, we propose to bring chain-of-thought reasoning into offline goal-conditioned RL, reformulating hierarchical control as the autoregressive sequential generation of latent subgoals and the primitive action within a single forward pass.

**MLP-Mixer** MLP-Mixer introduced a minimalist all-MLP backbone for vision by alternately applying token-mixing and channel-mixing MLPs to patch embeddings, achieving competitive classification performance without convolutions or attention (Tolstikhin et al. (2021)). Subsequent extensions have applied the same principles beyond standard imaging: dynamic token mixing for adaptive vision models (Wang et al. (2022b)), and fully MLP-based architectures for multivariate time series forecasting (Chen et al. (2023); Cho & Lee; Wang et al. (2024)). These advances underscore the MLP-Mixer's linear scaling and representational flexibility across modalities. To the best of our knowledge, CoGHP is the first work to adapt MLP-Mixer for offline goal-conditioned RL, enabling unified autoregressive sequence generation of hierarchical subgoals within a single end-to-end framework.

## 3 PROBLEM FORMULATION AND PRELIMINARIES

### 3.1 PROBLEM FORMULATION

We study offline goal-conditioned reinforcement learning in a Markov decision process $M = (\mathcal{S}, \mathcal{A}, P, r, \gamma, \rho_0)$, where $\mathcal{S}$ is the state space, $\mathcal{A}$ denotes the action space, $P(s' \mid s, a)$ denotes the transition dynamics, $r(s, g)$ denotes a reward function measuring progress toward goal $g$, $\gamma \in [0, 1)$ denotes the discount factor, and $\rho_0$ denotes the initial state distribution. A static dataset $\mathcal{D} = \{\tau_i\}_{i=1}^N$ of trajectories $\tau = (s_0, a_0, s_1, a_1, \ldots, s_T)$ is collected beforehand, and no further environment interaction is permitted. We assume a goal space $\mathcal{G} = \mathcal{S}$, and at evaluation time, each episode is paired with a goal $g \sim p(g)$. The objective is to learn a stationary policy $\pi_\theta(a \mid s, g)$ that maximizes the expected discounted return $J(\pi_\theta) = \mathbb{E}_{g \sim p(g), \tau \sim \mathcal{D}}[\sum_{t=0}^T \gamma^t r(s_t, g)]$.

### 3.2 GOAL-CONDITIONED IMPLICIT Q-LEARNING (IQL)

Implicit Q-Learning (IQL) (Kostrikov et al. (2021)) stabilizes offline RL by avoiding queries to out-of-distribution (OOD) actions through two key components: a state-value function $V_\psi(s)$ and an action-value function $Q_\theta(s, a)$. The value functions are trained via:

$$\mathcal{L}_Q(\theta) = \mathbb{E}_{(s,a,s') \sim \mathcal{D}} \left[ (r(s, a) + \gamma V_\psi(s') - Q_\theta(s, a))^2 \right], \tag{1}$$

$$\mathcal{L}_V(\psi) = \mathbb{E}_{(s,a)\sim\mathcal{D}} \left[ L_2^\tau \left( Q_{\bar{\theta}}(s,a) - V_\psi(s) \right) \right], \tag{2}$$

where $L_2^\tau(x) = |\tau - \mathbb{1}(x < 0)|x^2$ and $\tau \in [0.5, 1)$ controls conservatism (higher $\tau$ prioritizes optimistic returns), and $\bar{\theta}$ is a parameters of the target Q network. The policy $\pi_\phi(a|s)$ is then extracted via advantage-weighted regression (AWR) (Peters & Schaal (2007); Wang et al. (2020)):

$$J_\pi(\phi) = \mathbb{E}_{(s,a)\sim\mathcal{D}} \left[ \exp\left(\beta \cdot A(s,a)\right) \log_\pi \phi(a|s) \right], \tag{3}$$

with $A(s,a) = Q_\theta(s,a) - V_\psi(s)$ .

For goal-conditioned RL, IQL is extended to learn a goal-conditioned state-value function $V_\psi(s, g)$, preserving IQL's key advantage of stable value learning without requiring explicit Q-function evaluations on out-of-distribution actions (Ghosh et al. (2023)):

$$\mathcal{L}_V(\psi) = \mathbb{E}_{(s,s',g)\sim\mathcal{D}} \left[ L_2^\tau \left( r(s,g) + \gamma V_{\bar{\psi}}(s',g) - V_\psi(s,g) \right) \right]. \tag{4}$$

The corresponding policy is trained via a variant of AWR, which reweights behavior actions by exponentiated estimates of the goal-conditioned advantage:

$$J_\pi(\phi) = \mathbb{E}_{(s,a,s')\sim\mathcal{D}, g\sim p(g)} \left[ \exp\left(\beta \cdot A(s,a,g)\right) \log \pi_\phi(a|s,g) \right], \tag{5}$$

where $A(s,a,g) \approx \gamma V_\psi(s',g) + r(s,g) - V_\psi(s,g)$. This advantage-weighted policy extraction ensures that the learned policy focuses on high-value actions relative to each specific goal.

### 3.3 MLP-Mixer

MLP-Mixer (Tolstikhin et al. (2021)) is a simple, all-MLP architecture that was originally introduced for image classification tasks such as ImageNet and CIFAR. It avoids both convolution and self-attention, instead relying on alternating multi-layer perceptron blocks over spatial and channel dimensions to achieve competitive visual recognition performance using only MLPs. In its implementation, an input image is first divided into fixed-size patches and linearly projected to a sequence of token embeddings. Each Mixer layer then interleaves two MLP sub-layers: a token-mixing MLP that operates across the patch dimension to exchange information between spatial locations, and a channel-mixing MLP that acts independently on each token's feature channels to capture per-location feature interactions. Both sub-layers are wrapped with layer normalization, residual connections, and pointwise nonlinearities (e.g., GELU).

## 4 PROPOSED METHOD

We present the **Chain-of-Goals Hierarchical Policy (CoGHP)**, a novel framework that brings chain-of-thought reasoning from LLMs to offline goal-conditioned RL. Our proposed approach addresses the fundamental limitations that plague most existing offline hierarchical RL methods: single subgoal constraints, loss of final goal awareness when high-level guidance is erroneous, and fragmented optimization across separate networks. Our key insight is to reformulate hierarchical control as a sequence generation problem, where the policy autoregressively generates a sequence of latent subgoals and the primitive action, all conditioned on both the current state and the goal state. This formulation preserves final goal awareness and enables end-to-end optimization across all decision stages. This section details our architectural design (Section 4.1), describes the forward pass mechanism (Section 4.2), presents the training objectives (Section 4.3), and outlines the training procedure (Section 4.4).

### 4.1 ARCHITECTURE DESIGN

To implement this sequence generation paradigm, we require an architecture that can efficiently process a sequence of embedded tokens (state, goal, latent subgoals, and action) while modeling dependencies between sequence elements. For such sequential processing requirements, Transformer

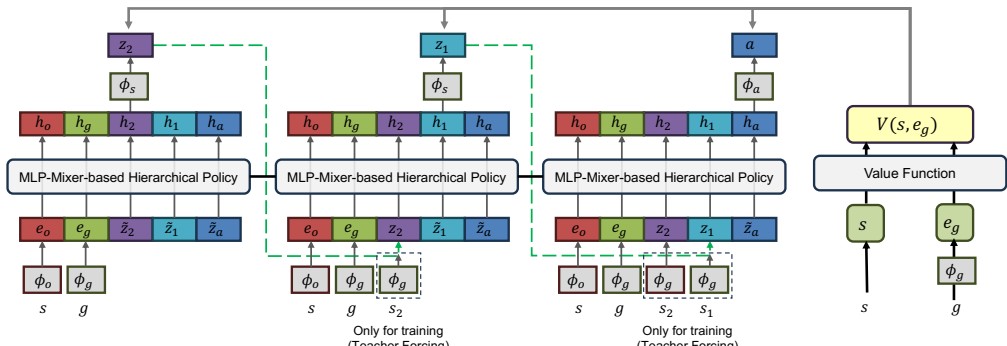

Figure 2: **Autoregressive Sequence Generation in CoGHP.** The policy autoregressively generates latent subgoals in order from most distant ($z_H$) to nearest ($z_1$) from the current state, and the primitive action $a$. At step $i$, the MLP-Mixer processes state embedding $e_o$, goal embedding $e_g$, previously generated subgoals, and remaining initial tokens to output $z_i$. This sequential generation ensures that each subgoal leverages information from all previously generated waypoints, enabling comprehensive hierarchical reasoning. During training, we apply teacher forcing by providing ground-truth subgoal embeddings to prevent error accumulation.

architectures are widely adopted across language and vision domains. However, while Transformers excel at capturing complex inter-element dependencies and dynamic interactions, they are less suited for settings where tokens have fixed position-dependent roles and the target signal primarily depends on its own temporal position rather than complex interactions Chen et al. (2023). Consequently, within our offline hierarchical RL framework, where each token position is assigned a fixed semantic role (such as current state, final goal, latent subgoal sequence, and primitive action), we found Transformer backbones to provide no clear generalization benefit and to exhibit reduced training stability in practice. We empirically verified this limitation in our experiments.

Our key architectural insight is to harness the MLP-Mixer architecture, which proves well-suited for sequence generation with position-dependent token roles. MLP-Mixer consists of alternating token-mixing and channel-mixing MLP layers that enable cross-token communication and per-token feature refinement using only simple feedforward operations. While MLP-Mixer is inherently sensitive to input token order and does not require separate positional embeddings, we augment it with a learnable causal token-mixer to better incorporate information from previously generated tokens during autoregressive subgoal and action generation. The causal mixer is implemented as a lower-triangular matrix applied to the stacked tokens, transforming each token into a weighted sum with previously generated tokens. This design enables better incorporation of sequential dependencies crucial for hierarchical control. Detailed specifications are provided in Appendix A.1.

## 4.2 FORWARD PASS AND SEQUENCE GENERATION

Our hierarchical policy first takes as input the token sequence $[e_o, e_g, \tilde{z}_H, \ldots, \tilde{z}_1, \tilde{z}_a]$, where $e_o = \phi_o(s)$ and $e_g = \phi_g(g)$ are state and goal embeddings, $\tilde{z}_{1:H}$ are learnable subgoal initial tokens, and $\tilde{z}_a$ is the action initial token. During the forward pass, these initial tokens are progressively filled in an autoregressive manner. At generation step $i$, the policy takes as input the state, the goal, all previously generated subgoals $z_{i+1:H}$, and the remaining initial tokens $\tilde{z}_{1:i}, \tilde{z}_a$. Importantly, we hypothesize that subgoals closer to the current state should incorporate more comprehensive information from the hierarchical reasoning process. Therefore, we configure our model to generate subgoals sequentially from those most distant from the current state ($z_H$) to those nearest ($z_1$). The MLP-Mixer backbone processes this sequence through token-mixer, causal mixer, and channel-mixer layers to output hidden state $h_i$, which is then passed through subgoal header $\phi_s$ to produce $z_i = \phi_s(h_i)$. After all $H$ latent subgoals are generated, the hidden state $h_a$ derived from the action initial token $\tilde{z}_a$ is passed through the action header $\phi_a$ to produce the primitive action $a = \phi_a(h_a)$. The complete sequence generation can be written as:

$$\pi_\theta(z_{1:H}, a|s, g) = \left( \prod_{i=H}^{1} \pi_\theta(z_i|e_o, e_g, z_{i+1:H}, \tilde{z}_{1:i}, \tilde{z}_a) \right) \pi_\theta(a|e_o, e_g, z_{1:H}, \tilde{z}_a), \qquad (6)$$

where the product notation $\prod_{i=H}^{1}$ indicates the generation order from $z_H$ to $z_1$, and $z_{i+1:H} = \emptyset$ when $i = H$. This autoregressive sequence generation mechanism is visualized in Figure 2.

### 4.3 TRAINING OBJECTIVES

To train this unified architecture, we propose an AWR-style objective that provides consistent training signals across all sequence elements. Our approach employs a shared value function to train both the latent subgoal sequence generation and final action prediction within the same network, ensuring coherent optimization across all hierarchical levels. While this shared value-based training strategy draws inspiration from HIQL (Park et al. (2023)), our key innovation lies in unifying all hierarchy levels within a single network architecture, contrasting with HIQL's approach of using separate network modules for different hierarchical components.

First, we learn $V_\psi(s, e_g)$ from the offline dataset $\mathcal{D}$ by minimizing the IQL temporal-difference error as defined in Equation 4. By training the value function on state-embedded goals, we can directly apply it to both embedded goals and latent subgoals, which reside in the same latent space. For details on value function training, please refer to Appendix A.2. Using advantage estimates derived from this value function, we define separate objectives for each prediction step. Our training objectives are:

$$J^{h_i}(\theta) = \mathbb{E}_{(s,g,s_{i:H}) \sim \mathcal{D}} \left[ \exp(\beta \cdot \tilde{A}^h(s, s_i, e_g)) \log \pi_\theta(z_i|e_o, e_g, z_{i+1:H}) \right], \qquad (7)$$

$$J^\ell(\theta) = \mathbb{E}_{(s,s',a,s_{1:H}) \sim \mathcal{D}} \left[ \exp(\beta \cdot \tilde{A}^\ell(s, a, z_H)) \log \pi_\theta(a|e_o, e_g, z_{1:H}) \right], \qquad (8)$$

where $\beta$ is a temperature parameter controlling the sharpness of advantage weighting. Following HIQL's advantage approximations, we use $\tilde{A}^h(s, s_i, e_g) \approx V_\psi(s_i, e_g) - V_\psi(s, e_g)$ and $\tilde{A}^\ell(s, a, z_H) \approx V_\psi(s', z_H) - V_\psi(s, z_H)$. The advantage terms quantify the value of each prediction step: $\tilde{A}^h(s, s_i, e_g)$ measures the benefit of reaching intermediate state $s_i$ toward goal $g$, while $\tilde{A}^\ell(s, a, z_H)$ evaluates action quality relative to the nearest generated subgoal. We extract target subgoals $s_{1:H}$ by sampling states at fixed $k$-step intervals along dataset trajectories, providing supervision for our latent subgoals $z_{1:H}$ to learn meaningful waypoint representations. This advantage weighting naturally guides the policy toward high-value subgoals and corresponding optimal actions.

These individual objectives are aggregated into a single end-to-end loss:

$$J_{\text{total}}(\theta) = \lambda_h \sum_{i=H}^{1} \gamma_h^{i-1} J^{h_i}(\theta) + \lambda_\ell J^\ell(\theta), \qquad (9)$$

where $\lambda_h$ and $\lambda_\ell$ are weight coefficients for the subgoal and action losses, respectively, and $\gamma_h$ is the discount factor that down-weights contributions from distant subgoals. The summation notation $\sum_{i=H}^{1}$ denotes the computation order from $J^{h_H}$ to $J^{h_1}$.

### 4.4 TRAINING PROCEDURE

CoGHP employs an alternating optimization scheme between the value function and the hierarchical policy. In each iteration, we first update $V_\psi$ using sampled transitions $(s, s', g)$, then fix the value function and train the policy $\pi_\theta$ using trajectory segments $(s, a, s', s_{1:H}, g)$. During policy training, we apply teacher forcing by providing ground-truth subgoal embeddings instead of using the policy's own predictions, preventing error accumulation during early training stages. Further details can be found in Appendix A.3.

For simplicity, we implement the generated latent subgoals as encoded future states. While the MLP-Mixer-based backbone of CoGHP can, in principle, accommodate alternative subgoal representations (e.g., learned skill primitives Pertsch et al. (2021) or abstract semantic embeddings Brohan et al. (2023)), such extensions would require additional dataset modalities or annotations and corresponding training objectives, and are therefore left to future work.

## 5 EXPERIMENTS

We conduct experiments to evaluate our approach through three comprehensive evaluations. First, we evaluate CoGHP's performance against strong baselines on challenging navigation and manipulation tasks. Second, we analyze the contribution of our architectural components, specifically the MLP-Mixer backbone and causal token-mixer, through ablation studies that reveal their increasing importance as task complexity grows. Third, we visualize the latent subgoal sequences generated by CoGHP to provide insights into how our chain-of-goals approach decomposes complex tasks.

### 5.1 EXPERIMENTAL SETUP

We evaluate CoGHP on the OGBench suite (Park et al. (2024)), a comprehensive benchmark for offline goal-conditioned RL that features diverse locomotion and manipulation environments (Figure 5). The navigation tasks include pointmaze, where a 2D point mass agent operates in a two-dimensional state-action space, and antmaze, which involves controlling a quadrupedal Ant agent with 8 degrees of freedom and more complex dynamics. We test across three environment sizes (medium, large, and giant) to assess how CoGHP's long-horizon reasoning capabilities scale with increasing maze complexity. For manipulation tasks, we focus on the cube and ccene environments to evaluate distinct aspects of object interaction. The cube task requires arranging blocks into target configurations through pick-and-place operations. We examine single, double, and triple cube variants to understand how performance scales with the number of objects requiring coordination. The scene environment presents a more sophisticated challenge, demanding multi-step sequential interactions such as unlocking, opening, placing, and closing operations in the correct order. This environment is particularly well-suited for evaluating long-horizon sequential reasoning and the agent's ability to handle diverse, structured object interactions. Following OGBench's evaluation protocol, we tested five predefined state-goal pairs per environment and reported the average success rate for all tasks.

We benchmark CoGHP against six representative algorithms from the OGBench reference suite. Goal-Conditioned Behavioral Cloning (GCBC) (Lynch et al. (2020); Ghosh et al. (2019)) formulates goal-conditioned control as a supervised learning problem by cloning the demonstrated action at each state–goal pair. Goal-Conditioned Implicit V-Learning (GCIVL) and Implicit Q-Learning (GCIQL) (Kostrikov et al. (2021)) both fit expectile-based value (and Q-value) estimators on offline data and then extract policies through advantage-weighted regression or behavior-constrained actor updates. Quasimetric RL (QRL) (Wang et al. (2023)) learns an asymmetric distance metric over states, enforcing the triangle inequality to induce a goal-conditioned value function. Contrastive RL (CRL) (Eysenbach et al. (2022)) uses a contrastive objective to train a Monte Carlo–style value estimator and performs a single-step policy improvement. Finally, Hierarchical Implicit Q-Learning (HIQL) (Park et al. (2023)) leverages a unified value function to derive separate high-level subgoal and low-level action policies via distinct advantage-weighted losses. Against these diverse baselines, we evaluate CoGHP's performance to demonstrate the effectiveness of our approach. Complete hyperparameter settings, including the number of latent subgoals generated by CoGHP, are provided in the Appendix B.

### 5.2 RESULTS ANALYSIS

**Navigation Performance** In the navigation benchmarks (pointmaze and antmaze in Table 1), CoGHP demonstrates superior performance across all task complexities, particularly excelling in the most challenging scenarios that require extensive multi-stage reasoning. On the giant maze variants, CoGHP achieved 79% on pointmaze-giant-navigate and 78% on antmaze-giant-navigate, significantly outperforming HIQL (46% and 65% respectively). This substantial performance gap highlights the limitations of HIQL's two-level hierarchical structure with separate networks when

Table 1: **Experimental Results on Navigation and Manipulation Environments.** Bold values indicate the highest performance values within a 5-point range of the maximum in each row. Standard deviations across 8 random seeds are shown. CoGHP achieves consistently superior or competitive results across diverse environments.

| Environment | Dataset | GCBC | GCIVL | GCIQL | QRL | CRL | HIQL | CoGHP (ours) |
|---|---|---|---|---|---|---|---|---|
| **pointmaze** | pointmaze-medium-navigate-v0 | $9 \pm 6$ | $63 \pm 6$ | $53 \pm 8$ | $82 \pm 5$ | $29 \pm 7$ | $79 \pm 5$ | $\mathbf{99} \pm 1$ |
| | pointmaze-large-navigate-v0 | $29 \pm 6$ | $45 \pm 5$ | $34 \pm 3$ | $86 \pm 9$ | $39 \pm 7$ | $58 \pm 5$ | $\mathbf{91} \pm 8$ |
| | pointmaze-giant-navigate-v0 | $1 \pm 2$ | $0 \pm 0$ | $0 \pm 0$ | $68 \pm 7$ | $27 \pm 10$ | $46 \pm 9$ | $\mathbf{79} \pm 8$ |
| **antmaze** | antmaze-medium-navigate-v0 | $29 \pm 4$ | $72 \pm 8$ | $71 \pm 4$ | $88 \pm 3$ | $\mathbf{95} \pm 1$ | $\mathbf{96} \pm 1$ | $\mathbf{97} \pm 2$ |
| | antmaze-large-navigate-v0 | $24 \pm 2$ | $16 \pm 5$ | $34 \pm 4$ | $75 \pm 6$ | $83 \pm 4$ | $\mathbf{91} \pm 2$ | $\mathbf{90} \pm 3$ |
| | antmaze-giant-navigate-v0 | $0 \pm 0$ | $0 \pm 0$ | $0 \pm 0$ | $14 \pm 3$ | $16 \pm 3$ | $65 \pm 5$ | $\mathbf{78} \pm 8$ |
| **cube** | cube-single-noisy-v0 | $8 \pm 3$ | $71 \pm 9$ | $\mathbf{99} \pm 1$ | $25 \pm 6$ | $38 \pm 2$ | $41 \pm 6$ | $\mathbf{97} \pm 3$ |
| | cube-double-noisy-v0 | $1 \pm 1$ | $14 \pm 3$ | $23 \pm 3$ | $3 \pm 1$ | $2 \pm 1$ | $2 \pm 1$ | $\mathbf{54} \pm 5$ |
| | cube-triple-noisy-v0 | $1 \pm 1$ | $9 \pm 1$ | $2 \pm 1$ | $1 \pm 0$ | $3 \pm 1$ | $2 \pm 1$ | $\mathbf{42} \pm 3$ |
| **scene** | scene-play-v0 | $5 \pm 1$ | $42 \pm 4$ | $51 \pm 4$ | $5 \pm 1$ | $19 \pm 2$ | $38 \pm 3$ | $\mathbf{78} \pm 7$ |

faced with tasks requiring coordination of multiple intermediate decisions. Unlike HIQL, which generates only a single intermediate subgoal, CoGHP's ability to perform multiple intermediate reasoning steps through its sequential subgoal chain enables more sophisticated navigation planning for complex maze environments.

**Manipulation Performance** CoGHP's advantages become even more pronounced in manipulation tasks (cube and scene in Table 1), where different types of sequential reasoning directly benefit from our unified sequence generation approach. Scene tasks require learning complex behavioral sequences, where agents must coordinate up to eight sequential atomic behaviors in the correct order. On scene task, CoGHP achieved 78% compared to HIQL's 38%, demonstrating how CoGHP's chain-of-goals approach enables proper decomposition of complex sequential tasks. Cube manipulation tasks present a different challenge, requiring repetitive pick-and-place operations where behavioral complexity is lower but precise motor control and correct placement ordering become critical. In these environments, HIQL exhibits performance degradation as the low-level policy lacks sufficient access to information about the final goal. This limitation becomes evident when comparing HIQL (41%) to GCIQL (99%) on cube-single, where HIQL's policy drift undermines the precise movements required for accurate cube placement. CoGHP addresses this fundamental issue through its unified optimization framework, achieving 97% on cube-single and maintaining strong performance even on the complex cube-triple (42%), where precise sequential placement of three cubes requires both accurate motor control and correct ordering strategies. This demonstrates CoGHP's ability to maintain awareness of the final goal while ensuring precise motor control, enabling successful manipulation across varying complexity levels.

## 5.3 ARCHITECTURAL COMPONENT ANALYSIS

To validate our architectural choices, we conducted ablation studies comparing CoGHP against two variants: a Transformer-based version (replacing MLP-Mixer blocks while keeping all other components identical) and an MLP-Mixer variant without a causal mixer. The results in Table 2 demonstrate that architectural advantages emerge progressively with task complexity. In simpler environments like antmaze-medium-navigate (97% for all variants), the choice of backbone architecture shows minimal impact, suggesting that basic sequential reasoning capabilities are sufficient. However, as task complexity increases, MLP-Mixer provides clear advantages over Transformers, with performance gaps widening substantially in challenging scenarios like cube-triple (42% vs 2%) and antmaze-giant-navigate (78% vs 66%). Similarly, the causal mixer component shows minimal contribution in simple tasks (antmaze-medium and cube-single), but becomes increasingly critical as the demands for hierarchical reasoning grow. In complex manipulation tasks requiring precise sequential coordination, the causal mixer provides substantial improvements (cube-triple: 42% vs 27%), confirming its critical role in enabling autoregressive generation where each subgoal can effectively incorporate information from previously generated tokens in the sequence. Further details on the Transformer baseline and its analysis are provided in Appendix B.3 and C.4.

Table 2: **Ablation Results on Architecture Variants.** Bold values indicate the highest performance values within a 5-point range of the maximum in each row. Standard deviations across 8 random seeds are shown. The experimental evidence shows that MLP-Mixer outperforms Transformer as an architectural foundation. Furthermore, the results highlight the important role of the causal mixer in achieving this performance.

| Environment | Transformer | CoGHP w/o causal mixer | CoGHP (Ours) |
|---|---|---|---|
| antmaze-medium-navigate-v0 | **97** $\pm 1$ | **97** $\pm 1$ | **97** $\pm 2$ |
| antmaze-giant-navigate-v0 | 66 $\pm 4$ | 71 $\pm 7$ | **78** $\pm 8$ |
| cube-single-noisy-v0 | 19 $\pm 2$ | **95** $\pm 4$ | **97** $\pm 3$ |
| cube-double-noisy-v0 | 11 $\pm 2$ | 44 $\pm 4$ | **54** $\pm 5$ |
| cube-triple-noisy-v0 | 2 $\pm 1$ | 27 $\pm 6$ | **42** $\pm 3$ |

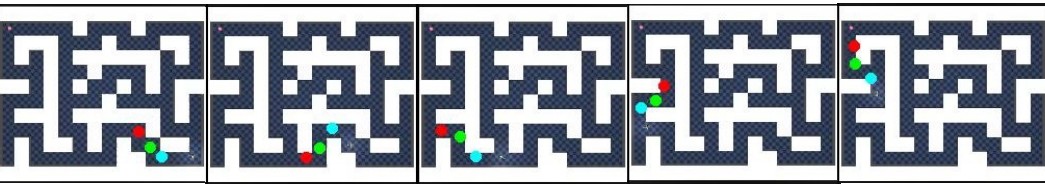

Figure 3: **Subgoal Visualization.** An agent located in the bottom-right corner is tasked with reaching the goal in the top-left. Here, the policy outputs three latent subgoals, plotted as blue, green, and red dots, ordered from nearest to farthest relative to the agent. The full version is in the Appendix C.10, and videos can be viewed in the supplementary materials.

## 5.4 SUBGOAL VISUALIZATIONS

We visualized latent subgoals in the antmaze-giant environment to examine how the chain of latent subgoals generated by CoGHP guides the agent. To map latent subgoals from the model's latent space back into the observation space, we added a subgoal decoder and trained it jointly with the hierarchical planner. During training, we applied an L2 loss between each decoded subgoal and its corresponding ground-truth subgoal from the dataset. For visualization, we extracted only the x and y coordinates of the decoded subgoals. The model was configured to generate a total of three subgoals. Figure 3 visualizes each subgoal. The fact that all three subgoals lie along or very close to the optimal path validates our hypothesis that CoGHP can effectively generate multiple intermediate goals to reach the final objective. Furthermore, this demonstrates that when generating the subgoal nearest to the current state, CoGHP integrates information from previously generated subgoals to help the agent produce optimal actions.

## 6 CONCLUSION

We introduced the Chain-of-Goals Hierarchical Policy (CoGHP), which brings the chain-of-thought reasoning paradigm from large language models into offline goal-conditioned RL. CoGHP tackles key limitations of earlier hierarchical methods, such as relying on a single intermediate subgoal, losing awareness of the final goal when subgoals are erroneous, and lacking end-to-end optimization. It does so by reformulating hierarchical control as autoregressive sequence generation within a unified framework. In a single forward pass, CoGHP generates a sequence of latent subgoals and the primitive action, with each subgoal serving as a "reasoning token." We pioneer the use of the MLP-Mixer architecture in hierarchical RL, enabling efficient cross-token communication and learning structural relationships that support hierarchical reasoning. Experiments on challenging navigation and manipulation benchmarks show that CoGHP consistently outperforms strong baselines, demonstrating its effectiveness for long-horizon offline control. Looking ahead, future work may explore adaptive mechanisms that adjust the number of subgoals based on task complexity and investigate more abstract forms of subgoal representation beyond encoded future states to further improve expressiveness and generalization.

## REPRODUCIBILITY STATEMENT

We implemented CoGHP based on the repository provided by OGBench (Park et al. (2024)). Training was performed on an NVIDIA GeForce RTX 3090 GPU, taking approximately 5 hours for state-based environments and about 15 hours for pixel-based environments. A detailed explanation of our framework's overall algorithm, model implementation details, and environment configurations are provided in Appendix A and Appendix B.

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

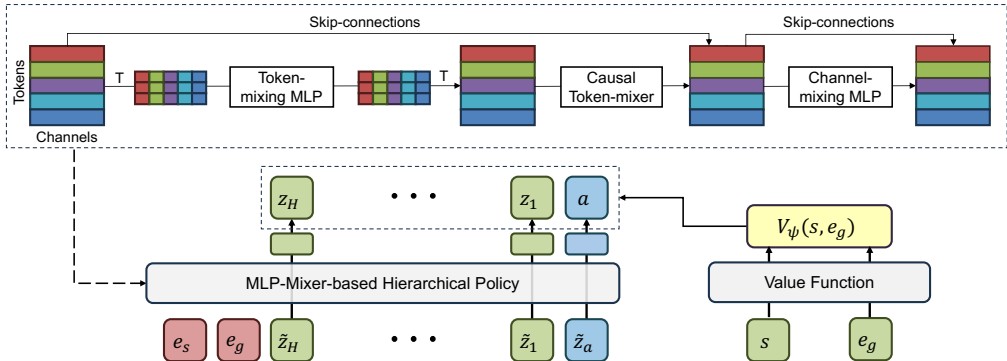

Figure 4: **CoGHP Architecture.** The framework comprises (1) a MLP-Mixer-based hierarchical policy that implements sequence generation for hierarchical control, autoregressively generating latent subgoals $z_H, \ldots, z_1$ ordered from farthest to nearest, and the primitive action $a$, and (2) a shared goal-conditioned value function $V_\psi(s, e_g)$ providing unified training signals for both subgoal generation and action prediction. The MLP-Mixer-based hierarchical policy takes H+3 tokens as input and processes them through alternating token-mixing, causal token-mixing, and channel-mixing layers to generate the output sequence.

## A ALGORITHMIC DETAILS

### A.1 ARCHITECTURE DETAILS

#### A.1.1 OVERVIEW

MLP-Mixer enables aggregation of both global and local features across tokens without complex attention. This ability to combine information from every token makes it an ideal backbone for our hierarchical policy, which must reason over an entire sequence of latent subgoals. Building on this insight, at each time step $t$, our hierarchical policy autoregressively generates a sequence of $H$ latent subgoals and one primitive action. During subgoal prediction, the CoGHP sequentially predicts latent subgoals from $z_H$, the furthest from the current state, toward $z_1$, the closest to the current state. Internally, our hierarchical policy maintains a fixed-length sequence of $T = 2+H+1$ tokens (two for the state and goal embeddings, $H$ for subgoal placeholders, and one for the action placeholder) and processes them through a single "Token-Mixer, Causal token-mixer, Channel-Mixer" block (Figure 4). This block shares parameters across all $H + 1$ prediction steps, enabling end-to-end gradient flow and parameter efficiency.

$$\text{input}_i = \begin{cases} (e_s, e_g, \tilde{z}_{1:H}, \tilde{z}_a) & i = H \\ (e_s, e_g, z_{i+1:H}, \tilde{z}_{1:i}, \tilde{z}_a) & i = H-1, \cdots, 1 \\ (e_s, e_g, z_{1:H}, \tilde{z}_a) & i = 0 \end{cases} \tag{10}$$

#### A.1.2 FORWARD PASS THROUGH THE MODIFIED MIXER

At each prediction step, the token sequence is first transposed and passed through the token-mixing MLP. It is then transposed back and multiplied by our learnable lower-triangular causal mixer, which enforces each token to integrate information from itself and all preceding tokens. The resulting token sequence is combined with the original input tokens via a skip connection. Next, it passes through the channel-mixing MLP, followed by another skip connection with the previous outputs. While both MLP blocks remain identical to those in the original Mixer, we introduce the causal mixer to impose sequential order and learnable inter-token dependencies.

Here, we illustrate with an example of four tokens $(e_o, e_g, z_1, z_a)$, showing how a learnable lower-triangular causal token-mixer is applied over them. First, let the token-mixing MLP produce per-token vectors

$$Y = \begin{pmatrix} y_1 \\ y_2 \\ y_3 \\ y_4 \end{pmatrix} \tag{11}$$

where $y_1 \leftrightarrow e_o,\ y_2 \leftrightarrow e_g,\ y_3 \leftrightarrow z_1,\ y_4 \leftrightarrow z_a$. We define a learnable $4 \times 4$ matrix

$$M = \begin{pmatrix} a_{11} & 0 & 0 & 0 \\ a_{21} & a_{22} & 0 & 0 \\ a_{31} & a_{32} & a_{33} & 0 \\ a_{41} & a_{42} & a_{43} & a_{44} \end{pmatrix}, \tag{12}$$

where each $a_{mn}$ (for $m \geq n$) is a trainable scalar and all entries above the diagonal are zero to block "future" tokens. We then compute the outputs as $Y' = M\,Y$, which component-wise yields

$$\begin{aligned}
y_1' &= a_{11}\,y_1, \\
y_2' &= a_{21}\,y_1 + a_{22}\,y_2, \\
y_3' &= a_{31}\,y_1 + a_{32}\,y_2 + a_{33}\,y_3, \\
y_4' &= a_{41}\,y_1 + a_{42}\,y_2 + a_{43}\,y_3 + a_{44}\,y_4.
\end{aligned} \tag{13}$$

Put simply, the lower-triangular causal mixer ensures that each token's representation is computed from its own features and those of all preceding tokens.

### A.2 TRAINING DETAILS

#### A.2.1 GOAL DISTRIBUTIONS

We use a mixture of three goal sources when training our value function. At each update, the goal $g$ is drawn with probability 0.2 from the current state $s_t$, with probability 0.5 from a future state sampled according to $\mathrm{Geom}(1 - \gamma)$, and with probability 0.3 from a uniformly chosen random state in the dataset. This combination balances learning from immediate rewards, long-horizon returns, and broad coverage of the state space. This sampling strategy follows approaches from Ghosh et al. (2023) and Park et al. (2023). We train the value function using these sampled states and goals via:

$$\mathcal{L}_V(\psi) = \mathbb{E}_{(s,s',g)\sim\mathcal{D}} \left[ L_2^\tau \left( r(s,g) + \gamma V_{\bar\psi}(s', \phi_{\psi_g}(g)) - V_\psi(s, \phi_{\psi_g}(g)) \right) \right]. \tag{14}$$

To generate training targets for CoGHP, we first sample a trajectory of length $T$ and pick a time index $t$. We uniformly sample the final goal $g$ from that trajectory. Next, we sample $H$ subgoals at fixed $k$-step intervals along the sampled trajectory. The $i$-th subgoal is the state $s_{\min(t+ik,\,T)}$, so that the policy sees subgoals starting from the farthest point $s_{\min(t+Hk,\,T)}$ and stepping inward by $k$ each prediction step until $s_{\min(t+k,\,T)}$.

#### A.2.2 GOAL REPRESENTATIONS

All tokens processed by the CoGHP MLP-Mixer backbone must share a common embedding dimension. We therefore set $d = 32$ for the maze navigation tasks and $d = 256$ for the Cube and Scene manipulation tasks. Each token is mapped to $\mathbb{R}^d$ via an encoder $\phi$ (state encoder $\phi_o$ or goal encoder $\phi_g$).

### A.3 ALGORITHM

Algorithm 1 provides a pseudocode for CoGHP.

---

**Algorithm 1** Chain-of-goals Hierarchical Planner (CoGHP)

---

**Require:** offline dataset $\mathcal{D}$
1: Initialize value function $V_\psi$, hierarchical policy $\pi_\theta$, state encoder $\phi_{\theta_o}$, goal encoder $\phi_{\psi_g}$, subgoal header $\phi_{\theta_s}$ and action header $\phi_{\theta_a}$     ($\{\psi_g\} \in \psi$, $\{\theta_o, \theta_s, \theta_a\} \in \theta$)
2: Initialize learning rates $\eta_\psi, \eta_\theta$
3: **for** each training iteration $n$ **do**
4:     Sample $(s, s', g) \sim \mathcal{D}$
5:     Compute loss $\mathcal{L}_V(\psi)$ using Equation 14
6:     $\psi \leftarrow \psi - \eta_\psi \nabla_\psi \mathcal{L}_V(\psi)$
7:     Sample $(s, a, s', s_{1:H}, g) \sim \mathcal{D}$
8:     **for** each subgoal prediction step $i = H, \ldots, 1$ **do**
9:         Predict latent subgoal $z_i = \pi_\theta \big( z_i \mid e_o, e_g, z_{i+1:H}, \tilde{z}_{1:i}, \tilde{z}_a \big)$
10:         Compute hierarchical objective $J^{h_i}(\theta)$ using Equation 7
11:     **end for**
12:     Predict primitive action $a = \pi_\theta \big( a \mid e_o, e_g, z_{1:H}, \tilde{z}_a \big)$
13:     Compute low-level objective $J^\ell(\theta)$ using Equation 8
14:     Compute total objective $J_{\text{total}}(\theta)$ using Equation 9
15:     $\theta \leftarrow \theta - \eta_\theta \nabla_\theta J_{\text{total}}(\theta)$
16: **end for**

---

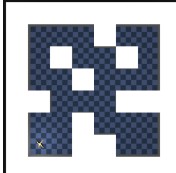 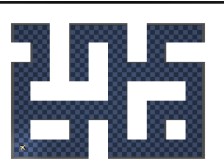 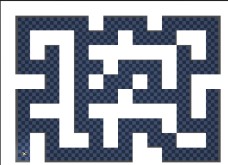 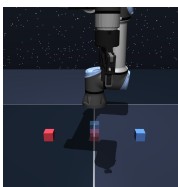 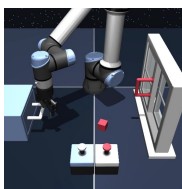

Figure 5: **Evaluation Environments.** Our experiments utilize environments from the OGBench suite: maze-medium, maze-large, and maze-giant (left to right) test navigation capabilities with increasing complexity using both Point mass and Ant agents; cube (fourth) evaluates block manipulation skills across single, double and triple cube variants; and scene (rightmost) examines multi-step interaction sequences requiring unlocking, opening, and manipulating objects.

# B    IMPLEMENTATION DETAILS

## B.1    ENVIRONMENT DETAILS

We evaluated CoGHP on a subset of OGBench (Park et al. (2024)) environments covering both navigation and manipulation challenges. For navigation, experiments took place in the pointmaze and antmaze domains, where the agent must traverse from a random start to a random goal within the mazes. Each domain includes medium, large, and giant variants to progressively test long-horizon reasoning. In pointmaze, a 2D point-mass agent operates in a two-dimensional state-action space, whereas antmaze uses the same maze layouts to challenge a quadrupedal ant agent with a 29-dimensional observation space and an 8-dimensional action space.

Manipulation tasks employ a 6-DoF UR5e arm with a Robotiq 2F-85 gripper in the cube and scene scenarios. In the cube environments, the agent arranges one to three cubes into a target configuration using pick-and-place, stacking, or swapping actions. Single-cube trials have a 28-dimensional observation space, double-cube trials have a 37-dimensional observation space, and triple-cube trials have a 46-dimensional observation space. All cube environments use a 5-dimensional action space corresponding to displacements in x position, y position, z position, gripper yaw, and gripper opening. The scene setup increases the observation space to 40 dimensions, which captures object poses and lock states, while retaining a 5-dimensional action space.

All environments use a sparse reward structure where the agent receives a reward of 0 upon successfully reaching the goal and -1 at each timestep when the goal has not been reached. Following OGBench's evaluation protocol, we tested five predefined state-goal pairs per environment and reported the average success rate for both navigation and manipulation tasks.

## B.2 HYPERPARAMETERS

We categorize our experimental environments into navigation, manipulation, and visual tasks, and summarize their hyperparameters in Table 3. For the navigation tasks, the values in {} denote the hyperparameters for the medium, large, and giant map sizes, respectively.

Table 3: CoGHP hyperparameters.

| Hyperparameter | Navigation | Manipulation | Visual-tasks |
|---|---|---|---|
| # gradient steps | 1000000 | 1000000 | 500000 |
| Batch size | | 256 | |
| Value MLP dimensions | | (512, 512, 512) | |
| Encoder MLP dimensions | | (512, 512, 512) | |
| Pixel-based Representation | | Impala CNN | |
| Header MLP dimensions | | (512, 512, 512) | |
| State/goal embedding dimensions | {32, 32, 128} | 256 | 32 |
| Token-mixer MLP dimensions | | (32, 32) | |
| Channel-mixer MLP dimensions | | (32, 32) | |
| # Subgoals $H$ | {1, 2, 2} | 1 | 1 |
| Weight coefficients $\lambda_h$ | {0.04, 0.02, 0.02} | 0.1 | 0.04 |
| Weight coefficients $\lambda_\ell$ | | 1 | |
| Subgoal discount factor $\gamma_h$ | | 0.8 | |
| subgoal step $k$ | {25, 50, 50} | 10 | 25 |
| Advantage temperature $\beta$ | | 3.0 | |
| Learning rate | | 0.0003 | |
| Nonlinearity | | GELU | |
| Optimizer | | Adam | |

## B.3 TRANSFORMER BASELINE

The Transformer baseline uses two layers, and the token dimension is matched to CoGHP's state-embedding dimension in all environments. The parameter counts are also closely aligned; for example, on antmaze-giant, the Transformer has 5.61M parameters and CoGHP has 5.54M parameters. Both models are trained with the same optimizer, learning rate schedule, and number of training steps. Under these settings, the Mixer–Transformer comparison in this work is fair and capacity-matched with respect to both model size and training configuration.

# C ADDITIONAL EXPERIMENTS

## C.1 PIXEL-BASED ENVIRONMENTS

We evaluated CoGHP on two additional OGBench benchmarks to test its versatility across pixel-based tasks. First, in visual-antmaze-medium, the agent receives only 64×64×3 RGB frames from a third-person perspective and must infer its position and orientation by parsing the maze floor's colored tiles rather than relying on raw coordinate inputs. This pixel-only task probes CoGHP's ability to learn robust visual representations and control under perceptual uncertainty. Second, the visual-cube environment follows the same visual setup as visual-antmaze, where the agent receives only 64×64×3 RGB frames from a third-person perspective. However, the manipulation arm is made transparent to ensure full observability of the object configurations and workspace.

Table 4 reports CoGHP's performance alongside six benchmark methods on the visual-antmaze-medium and visual-cube-single tasks. On visual-antmaze-medium, CoGHP achieves 95% average success while CRL and HIQL attain 94% and 93% respectively, demonstrating that CoGHP retains robust goal-conditioned control under pure pixel observations. On visual-cube-single, CoGHP achieves a 98% success rate, comparable to HIQL's 99% performance and significantly outperforming other methods. These results demonstrate that CoGHP can effectively extend to tasks requiring pixel-based observations, maintaining its advantages in both navigation and manipulation tasks under visual input constraints.

Table 4: **Experimental results on pixel-based environments.** Bold values indicate the highest performance values within a 5-point range of the maximum in each column. Standard deviations across 4 random seeds are shown.

| Algorithm | visual-antmaze-medium-navigate-v0 | visual-cube-single-noisy-v0 |
|---|---|---|
| GCBC | $11 \pm 2$ | $14 \pm 3$ |
| GCIVL | $22 \pm 2$ | $75 \pm 3$ |
| GCIQL | $11 \pm 1$ | $48 \pm 3$ |
| QRL | $0 \pm 0$ | $10 \pm 5$ |
| CRL | $\mathbf{94} \pm 1$ | $39 \pm 30$ |
| HIQL | $93 \pm 1$ | $\mathbf{99} \pm 0$ |
| CoGHP (ours) | $\mathbf{95} \pm 2$ | $\mathbf{98} \pm 1$ |

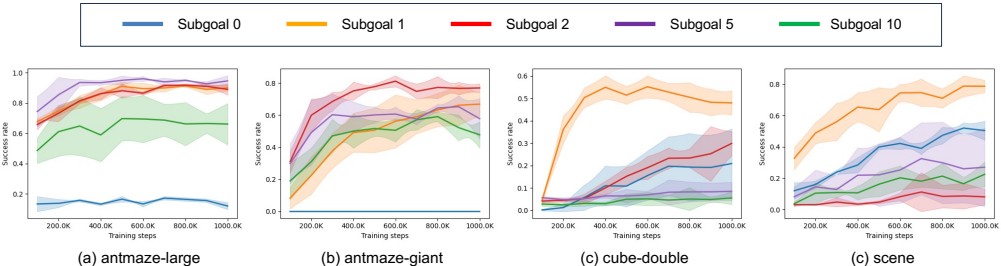

Figure 6: **CoGHP Architecture.** We compare the impact of the number of subgoals in our framework (with 8 different random seeds). Dark lines represent the average returns, and shaded areas represent standard deviations.

## C.2 SUBGOAL COUNT ANALYSIS

To investigate the impact of subgoal count for different task types, we conducted systematic experiments varying the subgoal count from 0 to 10 across representative navigation and manipulation environments. Figure 6 presents the performance comparison across antmaze-large, antmaze-giant, cube-double, and scene environments with subgoal counts of 0, 1, 2, 5, and 10.

In navigation tasks, subgoal generation proves essential for task completion. When no subgoals were generated ($H = 0$), both antmaze environments exhibited near-zero success rates, demonstrating the critical importance of hierarchical decomposition for long-horizon navigation. For antmaze-large, performance remained consistently high with 1, 2, and 5 subgoals, while increasing to 10 subgoals resulted in noticeable performance degradation. The antmaze-giant environment, being more complex, showed optimal performance with 2 subgoals, with all other settings (1, 5, and 10 subgoals) yielding inferior results. These findings indicate that while the exact number of subgoals affects performance, the presence of intermediate waypoints is crucial for successful navigation in complex maze environments.

Manipulation tasks revealed distinctly different patterns compared to navigation. Both cube-double and scene environments achieved optimal performance with a single subgoal ($H = 1$). Notably, unlike navigation tasks, these manipulation environments maintained reasonable performance even without subgoal generation ($H = 0$). However, generating more than one subgoal consistently degraded performance, indicating that excessive hierarchical decomposition can interfere with the precise control required for manipulation tasks. This suggests that in domains requiring fine motor control, multiple intermediate subgoals may introduce unnecessary complexity that hampers rather than helps task execution.

## C.3 SENSITIVITY ANALYSIS OF JOINT VARIATION OF SUBGOAL COUNT AND SUBGOAL STEP

To analyze how the subgoal configuration influences performance, we conducted a sensitivity study where the number of generated subgoals $H$ and the subgoal step $k$ were varied jointly. We evaluated $H \in \{1, 2, 5\}$; for navigation tasks we set $k \in \{10, 50, 100\}$, and for manipulation tasks we set

Table 5: **Sensitivity Analysis of Joint Variation of $H$ and $k$.** Success rate (%) with standard deviation across 8 seeds.

(a) antmaze-large-navigate-v0

|  | $k = 10$ | $k = 50$ | $k = 100$ |
|---|---|---|---|
| $H = 1$ | $59 \pm 11$ | $\mathbf{90} \pm 1$ | $86 \pm 3$ |
| $H = 2$ | $61 \pm 4$ | $\mathbf{90} \pm 3$ | $92 \pm 1$ |
| $H = 5$ | $41 \pm 4$ | $\mathbf{92} \pm 1$ | $81 \pm 4$ |

(b) antmaze-giant-navigate-v0

|  | $k = 10$ | $k = 50$ | $k = 100$ |
|---|---|---|---|
| $H = 1$ | $10 \pm 2$ | $66 \pm 7$ | $44 \pm 10$ |
| $H = 2$ | $32 \pm 7$ | $\mathbf{78} \pm 8$ | $44 \pm 5$ |
| $H = 5$ | $33 \pm 10$ | $61 \pm 7$ | $53 \pm 6$ |

(c) cube-double-noisy-v0

|  | $k = 5$ | $k = 10$ | $k = 20$ |
|---|---|---|---|
| $H = 1$ | $\mathbf{53} \pm 11$ | $\mathbf{54} \pm 5$ | $32 \pm 4$ |
| $H = 2$ | $16 \pm 4$ | $27 \pm 5$ | $8 \pm 2$ |
| $H = 5$ | $8 \pm 6$ | $5 \pm 4$ | $7 \pm 3$ |

(d) scene-play-v0

|  | $k = 5$ | $k = 10$ | $k = 20$ |
|---|---|---|---|
| $H = 1$ | $\mathbf{78} \pm 5$ | $\mathbf{78} \pm 7$ | $\mathbf{74} \pm 7$ |
| $H = 2$ | $1 \pm 1$ | $8 \pm 7$ | $50 \pm 9$ |
| $H = 5$ | $30 \pm 8$ | $29 \pm 15$ | $23 \pm 11$ |

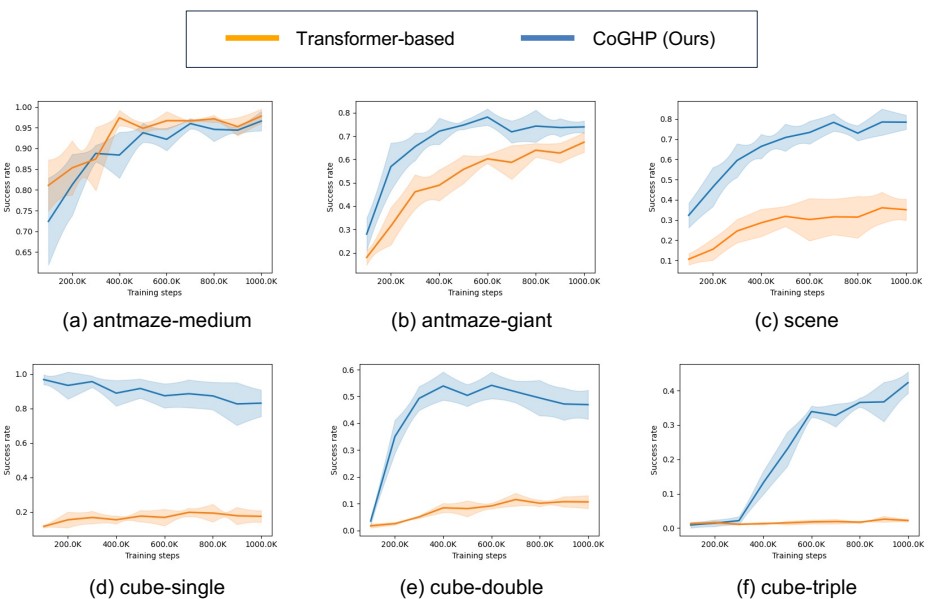

Figure 7: **Training curves for transformer baseline and CoGHP.** Dark lines represent the average returns, and shaded areas represent standard deviations across 8 random seeds.

$k \in \{5, 10, 20\}$. Table 5 summarizes the results. The navigation tasks are more sensitive to the spacing $k$ between subgoals than to the number of subgoals $H$, whereas the manipulation tasks are more sensitive to $H$. We interpret these differences as arising from task-specific characteristics. In navigation tasks, subgoals mainly serve as coarse waypoints that indicate intermediate directions or positions, so as long as they are spaced reasonably, performance is not highly sensitive to the exact number of subgoals. By contrast, in manipulation tasks, which require more precise control, an overly fine-grained subgoal chain can overconstrain the low-level policy and hinder accuracy. When $H = 1$ and the subgoal spacing is kept around $\{5, 10, 20\}$, however, performance is relatively less sensitive to $k$. This indicates that subgoal design interacts with task characteristics and supports the point made in the Section D that developing methods that are robust to the choice of subgoal horizon, or can automatically select an appropriate subgoal horizon for each task, is an important direction for future work.

Table 6: **Ablation Results on Subgoal Generation Order and Horizon.** Performance is reported as success rate (%) with standard deviation across 8 random seeds.

| Environment | forward (H=2) | forward (H=5) | reverse (H=2) | reverse (H=5) |
| --- | --- | --- | --- | --- |
| antmaze-large-navigate-v0 | 90 $\pm$ 2 | 92 $\pm$ 2 | 90 $\pm$ 3 | 92 $\pm$ 2 |
| antmaze-giant-navigate-v0 | 71 $\pm$ 2 | 50 $\pm$ 5 | 78 $\pm$ 8 | 61 $\pm$ 7 |

Table 7: **Ablation Results on Causal Mixer Variants.** Performance is reported as success rate (%) with standard deviation across 8 random seeds.

| Environment | w/o causal mixer | fixed causal mixer | CoGHP (Ours) |
| --- | --- | --- | --- |
| antmaze-medium-navigate-v0 | **97** $\pm$ 1 | **97** $\pm$ 1 | **97** $\pm$ 2 |
| antmaze-giant-navigate-v0 | 71 $\pm$ 7 | 72 $\pm$ 1 | **78** $\pm$ 8 |
| cube-single-noisy-v0 | 95 $\pm$ 4 | **98** $\pm$ 2 | 97 $\pm$ 3 |
| cube-double-noisy-v0 | 44 $\pm$ 4 | 51 $\pm$ 8 | **54** $\pm$ 5 |
| cube-triple-noisy-v0 | 27 $\pm$ 6 | 27 $\pm$ 4 | **42** $\pm$ 3 |

## C.4 TRANSFORMER-BASELINE ANALYSIS

As shown in Section 5.3 and Figure 7, CoGHP achieves higher performance than the transformer-based baseline in most environments. We interpret the performance difference between the Transformer baseline and CoGHP as largely stemming from how each architecture handles position-dependent tokens. In CoGHP, unlike text in LLMs where tokens have context-dependent meanings and roles, the input sequence is composed of structured position-dependent token roles, where each index has a fixed semantic function as "current state, final goal, sequential intermediate subgoals, and primitive action." In such settings, prior time-series studies (Zeng et al. (2023); Chen et al. (2023)) have observed that when the underlying signal is governed mainly by fixed position-dependent structure rather than rich context-dependent interactions across covariates, multivariate Transformer models can suffer from overfitting and degraded generalization, whereas time-step-dependent linear or MLP-based models tend to remain more robust. These results suggest that when token roles are relatively fixed and the signal is primarily position-dependent, the additional flexibility of data-dependent self-attention does not necessarily yield better generalization, making an MLP-Mixer backbone a natural architectural choice. Since the token roles are clearly fixed in CoGHP, this structural property helps explain the empirical Mixer-Transformer performance gap (Table 2).

## C.5 SUBGOAL GENERATION ORDER

Our initial design assumed that subgoals closer to the current state should aggregate more comprehensive information from the hierarchical reasoning process, and we therefore generated subgoals from the one farthest from the current state to the one closest to it. To test this assumption, we add an ablation that compares forward-order generation, which generates from the subgoal closest to the current state to the farthest subgoal, against reverse-order generation. We conduct experiments on navigation tasks where generating multiple subgoals yields stable performance, and examine how environment difficulty and the number of generated subgoals $H$ affect the impact of the subgoal generation order. The results are presented in Table 6. In the easier antmaze-large environment, forward and reverse generation perform similarly. In contrast, in antmaze-giant, reverse generation consistently outperforms forward generation, and the performance gap widens as $H$ increases. These findings support the validity of our initial design choice of generating subgoals in reverse order.

## C.6 ABLATION ON CAUSAL MIXER VARIANTS

To isolate the effect of the learnable causal mixer, we ran an ablation that compares (i) a variant that completely removes the causal mixer, (ii) a non-learnable causal mixer that replaces the learnable weights with fixed lower-triangular averaging, and (iii) default CoGHP with a learnable causal mixer

Table 8: **Loss-Weight Coefficient Sensitivity.** Performance is reported as success rate (%) with standard deviation across 4 random seeds.

| Environment / Dataset | 10 | 1 | 0.1 | 0.02 | 0.01 |
|---|---|---|---|---|---|
| antmaze-giant-v0-navigate | $0 \pm 0$ | $2 \pm 1$ | $66 \pm 4$ | $79 \pm 8$ | $65 \pm 4$ |
| cube-double-noisy-v0 | $52 \pm 4$ | $51 \pm 1$ | $54 \pm 5$ | $55 \pm 9$ | $56 \pm 7$ |

Table 9: **Teacher-Forcing Ablation.** Performance is reported as success rate (%) with standard deviation across 8 random seeds.

| Environment / Dataset | w/o Teacher Forcing | CoGHP (ours) |
|---|---|---|
| antmaze-giant-navigate-v0 | $18 \pm 5$ | $78 \pm 8$ |
| cube-double-noisy-v0 | $3 \pm 2$ | $54 \pm 5$ |

(6). In most environments, (i) removing the causal mixer and (ii) fixed lower-triangular averaging yield similar performance to each other, and both consistently underperform (iii) with the learnable causal mixer. This gap becomes more pronounced as task complexity increases. Thus, this experiment shows that simple causal masking or fixed averaging is not sufficient. It also indicates that a learnable causal mixer that learns the weights over past reasoning tokens plays a meaningful role in improving performance, especially on complex long-horizon tasks.

## C.7 LOSS-WEIGHT COEFFICIENT SENSITIVITY

To analyze the sensitivity of our method to hyperparameter choices, we conducted comparison experiments examining the impact of the loss-weight coefficient $\lambda_h$ across different task complexities. Our analysis reveals that $\lambda_h$, which scales the sub-goal generation term in Equation 9, influences training stability and performance. With a single predicted sub-goal ($H = 1$, e.g., cube-double), performance is stable over a wide span of $\lambda_h$, but when two sub-goals are generated ($H = 2$, e.g., antmaze-giant), setting $\lambda_h$ too high rapidly destabilises training and drives success toward zero. Accordingly, we keep $\lambda_\ell = 1$, hold $\gamma_h$ fixed, and tune $\lambda_h$ around the heuristic value $1/k$, which balances credit assignment across the latent chain while avoiding the sharp degradation observed at larger values.

## C.8 TEACHER FORCING ABLATION

To assess the effect of teacher forcing, we conduct an ablation that compares training with teacher forcing against training without teacher forcing under identical settings. The results are reported in Table 9. When the policy is trained without teacher forcing and rolled out using its own predicted subgoals, the success rate consistently decreases, indicating that standard teacher forcing plays an important role in achieving stable training and robust long-horizon rollouts in the CoGHP architecture.

## C.9 ADVANTAGE TEMPERATURE ABLATION

To study the sensitivity to the advantage temperature $\beta$, we conduct an ablation over $\beta \in \{1, 3, 10\}$. The results are reported in Table 10. On antmaze-giant, $\beta = 3$ achieves the highest success rate, with a mild drop at $\beta = 1$ and a larger decrease at $\beta = 10$. On cube-double, $\beta = 3$ again performs best, while $\beta = 1$ and $\beta = 10$ yield slightly lower but comparable success rates. Overall, these trends indicate that while $\beta = 3$ is a good default choice, CoGHP remains reasonably robust to the specific value of the advantage temperature within this range.

## C.10 SUBGOAL VISUALIZATIONS

This subsection presents an extended version of the subgoal visualization analysis from the main text. This extended sequence offers insight into how CoGHP's autoregressive subgoal generation guides the agent through complex navigation tasks. In the antmaze-giant environment, we decode

Table 10: **Advantage Temperature Sensitivity.** Performance is reported as success rate (%) with standard deviation across 8 random seeds.

| Environment / Dataset | 1 | 3 | 10 |
|---|---|---|---|
| antmaze-giant-navigate-v0 | 75 ± 2 | 78 ± 8 | 61 ± 6 |
| cube-double-noisy-v0 | 47 ± 2 | 54 ± 5 | 50 ± 4 |

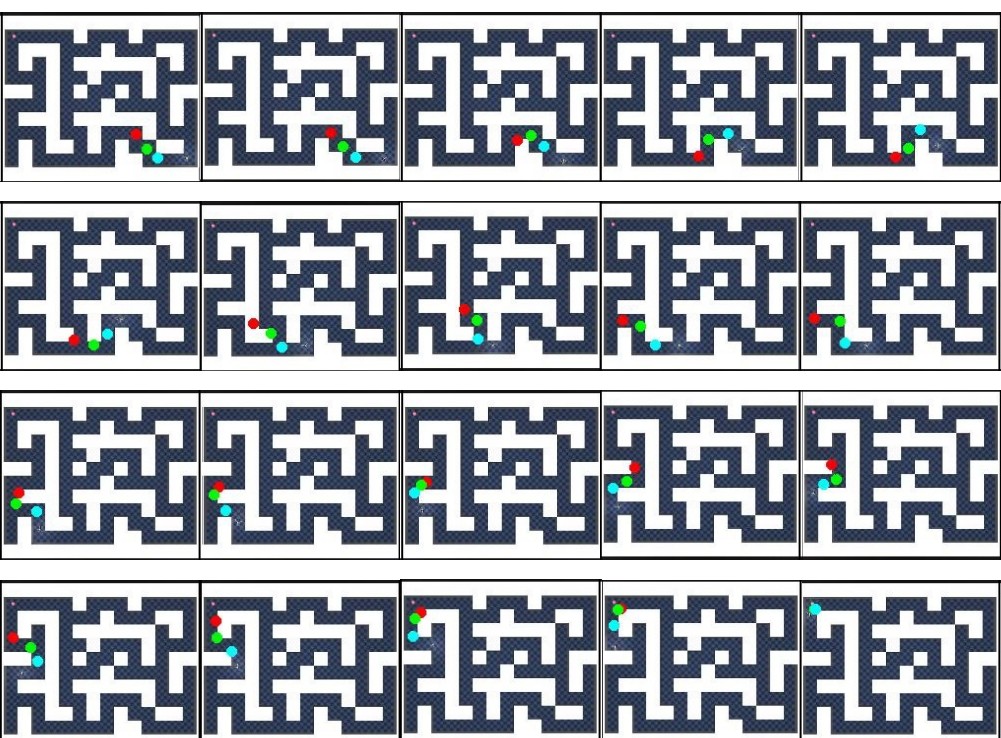

Figure 8: **AntMaze Subgoal Visualization.** To examine the role of CoGHP's latent subgoal chain, we decoded and visualized these subgoals in the antmaze-giant environment. In this scenario, the agent starts in the bottom-right corner and must reach the goal in the top-left. For this example, we configured the policy to output three latent subgoals, which we plotted as colored dots in blue, green, and red, ordered from nearest to farthest from the agent.

multiple latent subgoal sequences into coordinates to examine how they are positioned and what roles they play, and in the pixel-based visual-antmaze environment, we decode latent subgoals into images to verify that CoGHP produces meaningful subgoals even under more complex observations.

In the antmaze-giant environment, the results (Figure 3) show that while the farthest subgoal (red dot) sometimes positions itself in unreachable locations such as walls, the nearest subgoal (blue dot) consistently provides the agent with a reliably accessible intermediate destination by considering previously generated subgoals as well as the final objective. Another notable observation is how subgoals are positioned when approaching the final destination. When the final goal is sufficiently distant from the agent, the generated subgoals maintain regular spacing intervals. However, as the agent approaches the final goal, we observe a phenomenon where the subgoals begin to overlap. This demonstrates that CoGHP does not rigidly adhere to maintaining fixed intervals between subgoals, but rather generates optimal subgoals specifically tailored for the agent to successfully reach the final goal.

For the visual antmaze environments, we decode latent subgoal embeddings into images for qualitative visualization. The decoder takes a latent vector as input, projects it with a fully connected layer into a small spatial feature map (e.g., $8 \times 8$ with multiple channels), and then applies a stack of transposed convolutions with stride 2 and ReLU activations to progressively upsample the features.

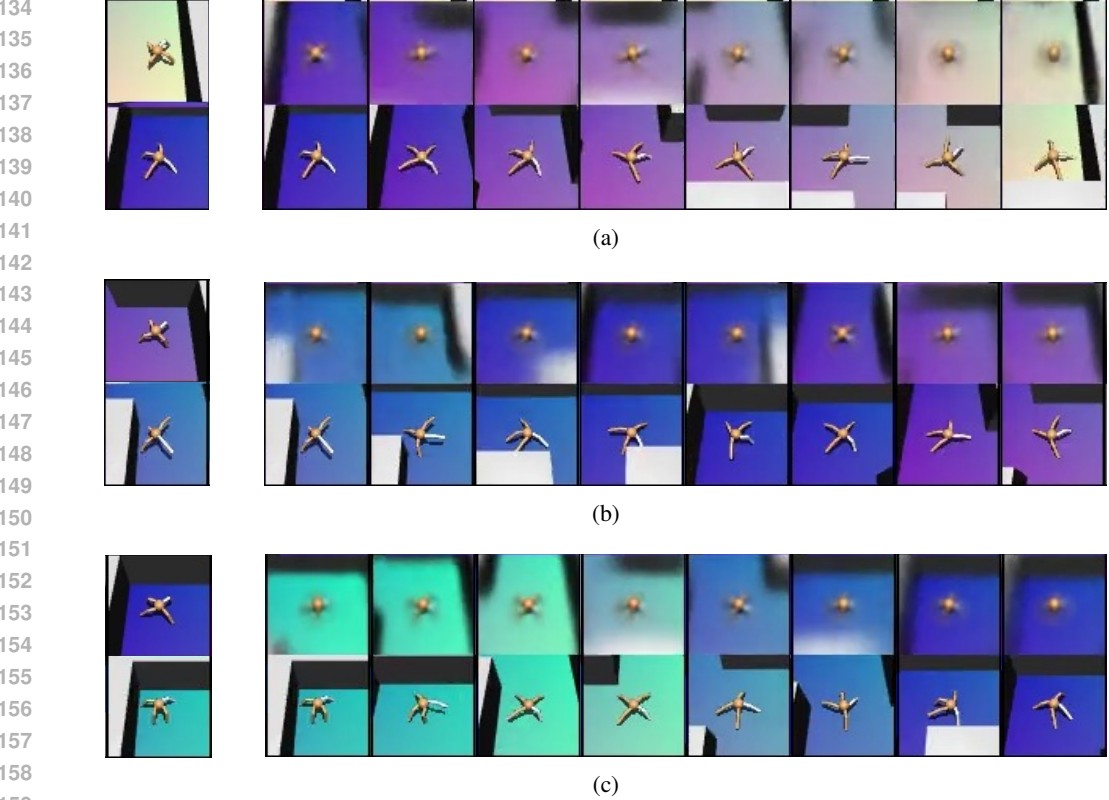

(a)

(b)

(c)

Figure 9: **Visual-AntMaze Subgoal Visualization.** We decoded and visualized the generated subgoals in the visual-antmaze environment. In the leftmost image, the lower panel shows the initial state, and the upper panel shows the goal state.

A final transposed convolution produces an image with the target number of channels, and a bilinear resize step is applied if necessary to match the exact target resolution. The decoder is trained with a simple reconstruction objective that combines mean-squared error (MSE) and $\ell_1$ loss between the decoded image and the target future-state image. As illustrated in Figure 8, CoGHP generates subgoals that still guide the agent toward the goal in this pixel-based setting. In visual-antmaze, the agent must infer its location from floor colors and wall layouts rather than explicit coordinates, and the decoded subgoal images reflect these cues by highlighting intermediate states that the agent should reach on the way to the goal. This shows that CoGHP can produce meaningful subgoals even when they must be expressed in a more complex image-based form rather than simple coordinate space.

### C.11 PER-TASK RESULTS

OGBench evaluates each dataset using five pre-defined evaluation tasks, each specified by a distinct initial state and goal state. These tasks can require qualitatively different behaviors to reach the corresponding goals. (For detailed descriptions of each individual task, please refer to the OGBench paper (Park et al. (2024)).) To examine performance at a finer granularity, we report per-task success rates in Tables 11, 12, 13, and 14. While certain individual tasks are better solved by specific baselines, CoGHP attains a higher overall average performance across the five tasks in most of the environments.

## D LIMITATIONS

Despite its effectiveness, CoGHP has several primary limitations. First, the number of latent subgoals and their timestep intervals during training must be predefined as hyperparameters. This can

Table 11: Per-task performance on PointMaze.

| Environment Type | Dataset | Task | GCBC | GCIVL | GCIQL | QRL | CRL | HIQL | CoGHP (ours) |
|---|---|---|---|---|---|---|---|---|---|
| pointmaze | pointmaze-medium-navigate-v0 | task1 | 30 ±27 | 88 ±16 | **97** ±4 | **100** ±0 | 20 ±6 | 99 ±1 | **100** ±0 |
| | | task2 | 3 ±2 | **95** ±10 | 76 ±29 | 94 ±17 | 45 ±25 | 87 ±7 | **100** ±0 |
| | | task3 | 5 ±5 | 37 ±28 | 10 ±28 | 23 ±20 | 30 ±4 | 55 ±13 | **95** ±4 |
| | | task4 | 0 ±1 | 2 ±2 | 0 ±0 | 94 ±14 | 28 ±29 | 82 ±12 | **100** ±0 |
| | | task5 | 4 ±3 | 92 ±7 | 79 ±6 | **97** ±8 | 24 ±13 | 70 ±10 | **100** ±0 |
| | | overall | 9 ±6 | 63 ±6 | 53 ±8 | 82 ±5 | 29 ±7 | 79 ±5 | **99** ±1 |
| | pointmaze-large-navigate-v0 | task1 | 63 ±11 | 76 ±23 | 86 ±14 | **95** ±8 | 42 ±27 | 83 ±13 | **100** ±0 |
| | | task2 | 1 ±2 | 0 ±0 | 0 ±0 | **100** ±0 | 31 ±24 | 2 ±7 | 54 ±40 |
| | | task3 | 10 ±7 | 98 ±5 | 83 ±8 | 40 ±50 | 78 ±7 | 88 ±10 | **100** ±0 |
| | | task4 | 20 ±18 | 0 ±0 | 0 ±0 | **96** ±7 | 24 ±14 | 72 ±19 | 95 ±5 |
| | | task5 | 52 ±17 | 53 ±20 | 0 ±0 | **96** ±7 | 20 ±10 | 46 ±16 | 99 ±1 |
| | | overall | 29 ±6 | 45 ±5 | 34 ±3 | **86** ±9 | 39 ±7 | 58 ±5 | 91 ±8 |
| | pointmaze-giant-navigate-v0 | task1 | 1 ±3 | 0 ±0 | 0 ±0 | 98 ±7 | 6 ±15 | 0 ±0 | **100** ±0 |
| | | task2 | 1 ±4 | 0 ±0 | 0 ±0 | **92** ±16 | 28 ±10 | 72 ±17 | 74 ±16 |
| | | task3 | 0 ±0 | 0 ±1 | 0 ±0 | 68 ±27 | 9 ±5 | 32 ±11 | **77** ±23 |
| | | task4 | 0 ±0 | 0 ±0 | 0 ±0 | 66 ±20 | 64 ±17 | 60 ±22 | **100** ±0 |
| | | task5 | 5 ±12 | 0 ±0 | 0 ±0 | 19 ±32 | 29 ±28 | **66** ±20 | 43 ±40 |
| | | overall | 1 ±2 | 0 ±0 | 0 ±0 | 68 ±7 | 27 ±10 | 46 ±9 | **79** ±8 |

Table 12: Per-task performance on AntMaze.

| Environment Type | Dataset | Task | GCBC | GCIVL | GCIQL | QRL | CRL | HIQL | CoGHP (ours) |
|---|---|---|---|---|---|---|---|---|---|
| antmaze | antmaze-medium-navigate-v0 | task1 | 35 ±9 | 81 ±10 | 63 ±9 | 93 ±2 | 97 ±1 | 94 ±2 | **98** ±2 |
| | | task2 | 21 ±7 | 85 ±5 | 78 ±8 | 90 ±5 | 95 ±2 | **97** ±1 | 96 ±2 |
| | | task3 | 24 ±6 | 60 ±13 | 71 ±8 | 86 ±6 | 92 ±3 | 96 ±2 | **99** ±1 |
| | | task4 | 28 ±7 | 42 ±25 | 59 ±12 | 83 ±4 | 94 ±5 | **96** ±2 | 95 ±2 |
| | | task5 | 37 ±10 | 92 ±3 | 85 ±7 | 88 ±8 | **96** ±2 | **96** ±2 | 95 ±3 |
| | | overall | 29 ±4 | 72 ±8 | 71 ±4 | 88 ±3 | 95 ±1 | 96 ±1 | **97** ±2 |
| | antmaze-large-navigate-v0 | task1 | 6 ±3 | 16 ±12 | 21 ±6 | 71 ±15 | 91 ±3 | **93** ±3 | **93** ±3 |
| | | task2 | 16 ±4 | 5 ±6 | 25 ±7 | 77 ±7 | 62 ±14 | 78 ±9 | **85** ±5 |
| | | task3 | 65 ±4 | 49 ±18 | 80 ±5 | 94 ±2 | 91 ±22 | **96** ±2 | **96** ±3 |
| | | task4 | 14 ±3 | 2 ±19 | 19 ±6 | 64 ±8 | 85 ±11 | **94** ±3 | 86 ±3 |
| | | task5 | 18 ±4 | 5 ±2 | 26 ±9 | 67 ±8 | 85 ±13 | **94** ±3 | 90 ±3 |
| | | overall | 24 ±2 | 16 ±5 | 34 ±4 | 75 ±6 | 83 ±4 | **91** ±2 | 90 ±3 |
| | antmaze-giant-navigate-v0 | task1 | 0 ±0 | 0 ±0 | 0 ±0 | 1 ±2 | 2 ±2 | 47 ±10 | **77** ±14 |
| | | task2 | 0 ±0 | 0 ±0 | 0 ±0 | 17 ±5 | 21 ±10 | 74 ±5 | **88** ±4 |
| | | task3 | 0 ±0 | 0 ±0 | 0 ±0 | 14 ±8 | 5 ±5 | 55 ±7 | **72** ±5 |
| | | task4 | 0 ±0 | 0 ±0 | 0 ±0 | 18 ±6 | 35 ±9 | 69 ±5 | **78** ±9 |
| | | task5 | 1 ±1 | 1 ±1 | 1 ±1 | 18 ±6 | 16 ±10 | **82** ±4 | 76 ±7 |
| | | overall | 0 ±0 | 0 ±0 | 0 ±0 | 14 ±3 | 16 ±3 | 65 ±5 | **78** ±4 |

lead to task under-decomposition or over-decomposition, making the agent's performance heavily dependent on these hyperparameter choices. Second, CoGHP may struggle when applied to environments or tasks that differ significantly from the training dataset or when confronting tasks not present in the original data. Third, the current implementation instantiates subgoals exclusively as encoded future states, even though the MLP-Mixer-based backbone of CoGHP could in principle accommodate alternative subgoal representations such as learned skill primitives or abstract semantic embeddings Pertsch et al. (2021); Brohan et al. (2023). Finally, the present work considers only unidirectional autoregressive generation of the subgoal sequence and does not explore alternative subgoal generation schemes.

To address the first limitation, future work could introduce adaptive mechanisms that dynamically adjust the number of subgoals or their temporal spacing based on task complexity. Alternatively, the system could be configured to generate a sufficient number of subgoals while enabling the agent to adaptively terminate subgoal generation and directly produce primitive actions when appropriate. For the second limitation, potential solutions include developing rapid adaptation techniques that enable agents trained in one environment to quickly generalize to new environments or tasks. Another promising direction involves training on diverse, multi-domain datasets rather than learning each task individually, thereby creating more generalizable models capable of broader application across different scenarios and environments. To mitigate the third limitation, future work could augment offline datasets with additional modalities or annotations and introduce training objectives that align alternative subgoal latents (e.g., skill or language embeddings) with the future state distribution, enabling CoGHP to operate with non-state subgoals. Concretely, using learned skill primitives as subgoals would entail first learning skill embeddings and a decoder from offline play data, then defining CoGHP's subgoal tokens as these skill latents, and finally training the shared value function over the state–goal–skill space. Likewise, using language-based semantic subgoals would require language annotations for each subgoal, a pretrained language encoder, and an additional objective

Table 13: Per-task performance on Cube.

| Environment Type | Dataset | Task | GCBC | GCIVL | GCIQL | QRL | CRL | HIQL | CoGHP (ours) |
|---|---|---|---|---|---|---|---|---|---|
| cube | cube-single-noisy-v0 | task1 | 5 ±3 | 71 ±12 | **100** ±1 | 17 ±12 | 39 ±4 | 48 ±6 | **97** ±2 |
| | | task2 | 7 ±5 | 70 ±11 | **100** ±0 | 23 ±8 | 39 ±7 | 39 ±7 | **96** ±2 |
| | | task3 | 1 ±1 | 67 ±10 | **99** ±1 | 4 ±3 | 36 ±7 | 41 ±7 | **97** ±3 |
| | | task4 | 16 ±5 | 76 ±10 | **98** ±2 | 47 ±14 | 36 ±4 | 36 ±5 | **96** ±2 |
| | | task5 | 12 ±5 | 70 ±12 | **100** ±0 | 37 ±9 | 42 ±5 | 44 ±9 | **97** ±3 |
| | | overall | 8 ±3 | 71 ±9 | **99** ±1 | 25 ±6 | 38 ±2 | 41 ±6 | **97** ±3 |
| | cube-double-noisy-v0 | task1 | 7 ±3 | 53 ±11 | 64 ±8 | 16 ±5 | 9 ±5 | 10 ±3 | **79** ±15 |
| | | task2 | 0 ±0 | 10 ±4 | 16 ±4 | 0 ±0 | 0 ±0 | 1 ±0 | **75** ±18 |
| | | task3 | 0 ±0 | 1 ±1 | 6 ±4 | 0 ±0 | 0 ±0 | 1 ±1 | **49** ±11 |
| | | task4 | 0 ±0 | 4 ±2 | 11 ±3 | 0 ±0 | 0 ±1 | 0 ±1 | **24** ±10 |
| | | task5 | 0 ±0 | 4 ±2 | 20 ±4 | 0 ±0 | 0 ±1 | 0 ±1 | **44** ±15 |
| | | overall | 1 ±1 | 14 ±3 | 23 ±3 | 3 ±1 | 2 ±2 | 2 ±1 | **54** ±5 |
| | cube-triple-noisy-v0 | task1 | 6 ±3 | 44 ±7 | 8 ±2 | 5 ±2 | 13 ±6 | 8 ±3 | **77** ±10 |
| | | task2 | 0 ±0 | 0 ±0 | 1 ±1 | 0 ±0 | 0 ±0 | 0 ±0 | **2** ±2 |
| | | task3 | 0 ±0 | 2 ±1 | 0 ±0 | 0 ±0 | 0 ±0 | 0 ±0 | **62** ±9 |
| | | task4 | 0 ±0 | 0 ±0 | 0 ±0 | 0 ±0 | 0 ±0 | 0 ±0 | **53** ±2 |
| | | task5 | 0 ±0 | 0 ±0 | 0 ±0 | 0 ±0 | 0 ±0 | 0 ±0 | **18** ±8 |
| | | overall | 1 ±1 | 9 ±1 | 2 ±1 | 1 ±0 | 3 ±1 | 2 ±1 | **42** ±3 |

Table 14: Per-task performance on Scene.

| Environment Type | Dataset | Task | GCBC | GCIVL | GCIQL | QRL | CRL | HIQL | CoGHP (ours) |
|---|---|---|---|---|---|---|---|---|---|
| scene | scene-play-v0 | task1 | 18 ±7 | 75 ±5 | **93** ±4 | 19 ±4 | 49 ±7 | 40 ±4 | **97** ±3 |
| | | task2 | 1 ±1 | 62 ±8 | 82 ±8 | 1 ±1 | 12 ±4 | 40 ±9 | **93** ±7 |
| | | task3 | 2 ±1 | 64 ±7 | 72 ±10 | 1 ±1 | 26 ±8 | 36 ±5 | **85** ±8 |
| | | task4 | 3 ±2 | 7 ±4 | 8 ±3 | 5 ±2 | 5 ±2 | 55 ±5 | **73** ±15 |
| | | task5 | 0 ±0 | 2 ±1 | 1 ±1 | 0 ±1 | 1 ±1 | 20 ±5 | **43** ±20 |
| | | overall | 5 ±1 | 42 ±4 | 51 ±4 | 5 ±1 | 19 ±2 | 38 ±3 | **78** ±7 |

to align language embeddings with the future state distribution. For the fourth limitation, a natural extension is to equip CoGHP with bidirectional or diffusion-style planners in the subgoal space that refine the entire subgoal sequence while retaining the simplicity and effectiveness of the current causal formulation.

# E    THE USE OF LARGE LANGUAGE MODELS (LLMS)

Large language models (LLMs) were used only for polishing and editing the language, such as improving sentence structure, grammar, and readability. All research activities of this work were conducted entirely by the authors.

