# OpenReview forum: "Chain-of-Goals Hierarchical Policy for Long-Horizon Offline Goal-Conditioned RL"
_ICLR.cc/2026/Conference — Submitted to ICLR 2026_

### Official Review · Reviewer_7rmc · 2025-10-30

**Soundness:** 3
**Presentation:** 2
**Contribution:** 3
**Rating:** 6
**Confidence:** 3

**Summary:**

The paper proposes Chain-of-Goals Hierarchical Policy (CoGHP), a framework for offline goal-conditioned RL. Instead of using separate high- and low-level networks, CoGHP generates multiple latent subgoals and the final action within a single forward pass. In particular, it adopts an MLP-Mixer backbone and demonstrates its advantage over transformer backbone. Experiments on navigation and manipulation benchmarks show that CoGHP consistently outperforms strong baselines like HIQL, especially on long-horizon tasks.

**Strengths:**

1. Reformulating hierarchical RL as autoregressive sequence generation within a single unified network is a neat and interesting idea that simplifies architecture design and enables end-to-end optimization.
2. The proposed CoGHP consistently outperforms strong baselines across both navigation and manipulation tasks.
3. The finding that an MLP-Mixer backbone outperforms the commonly used Transformer architecture is interesting.

**Weaknesses:**

1. The connection with chain-of-thought reasoning from LLMs is weak and a stretch. It is not convincing to consider the subgoal token as reasoning token. Framing offline RL as a sequence modeling problem has a well established literature. It is more appropriate to discuss the connection with this line of work such as DecisionTransformer and Trajectory Transformer.
2. The authors argue that generating the subgoal sequence in a reverse order is better. However, there is no corresponding ablation study to support that claim. Also, ideally, the subgoal sequence should be optimized jointly. In this sense, alternative formulations such as bidirectional generation or iterative refinement (e.g., diffusion-based planning) could be more principled.
3. The authors claim that the framework can handle other forms of subgoal representation. However, this generality seems non-trivial. Incorporating abstract or semantic subgoals would likely require additional modalities or dataset features. The authors should further elaborate on the additional modifications or assumptions needed to make such accommodations.

**Questions:**

1. The empirical results clearly show that the MLP-Mixer backbone outperforms the Transformer variant. Could the authors further elaborate on why this is the case? Which aspects of the MLP-Mixer architecture contribute to this advantage? In particular, could the authors clarify what is meant by “stepwise procedural consistency", and why is this property important for hierarchical RL in the proposed framework?
2. Does the proposed framework reduce to HIQL when the planning horizon is one? An ablation on the number of predicted subgoals (planning horizon) would help clarify how much of the performance gain arises from multi-step subgoal prediction versus the unified architecture.

---

> ### Author Response · Authors · 2025-11-21
> **Response to Reviewer 7rmc (1/4)**
>
> We sincerely thank you for your valuable feedback on our work. Below are our responses to the feedback provided by the reviewer 7rmc.
>
> > **W1: The connection with chain-of-thought reasoning from LLMs is weak and a stretch.**
>
> Thank you for connecting CoGHP to the “offline RL as sequence modeling” literature, including Decision Transformer and Trajectory Transformer. We agree with the reviewer that CoGHP is related to these methods through its use of a sequence-based architecture. However, there is a difference in the role played by this sequence modeling. Decision Transformer and Trajectory Transformer encode offline trajectory as a single long sequence of (returns-to-go or rewards, s, a) and use it for return-conditioned action prediction (Decision Transformer) or beam-search-based planning (Trajectory Transformer). In contrast, CoGHP defines, at each decision step, a structured sequence of “current state, final goal, latent subgoal chain, and action” and performs hierarchical subgoal inference and action generation on this sequence.
>
> For this reason, we view CoGHP as more closely aligned with the chain-of-thought notion of “thinking step by step.” More specifically, we focused on (i) unfolding a complex problem into a sequence of intermediate tokens, (ii) keeping this sequence conditional on the original query while passing through intermediate steps to produce the final output, and (iii) handling the entire process within a single unified sequence model. CoGHP builds on this perspective to mitigate the limitations of prior offline hierarchical RL methods that rely on two separately trained networks. Specifically, it (i) unfolds goals that are difficult to solve with a direct state–action mapping into a sequence of latent subgoal tokens, (ii) keeps this sequence conditional on the final goal while generating primitive actions through these intermediate subgoals, and (iii) processes the whole hierarchy end-to-end within a single unified model. This chain-of-thought-style “input - intermediate reasoning steps - primitive action” structure is consistent with recent works on robotic control with vision-language-action models (VLA), such as ECoT [1] and CoT-VLA [2], in which the model first produces intermediate embodied or visual reasoning steps and then predicts actions. In this sense, our work inherits the step-by-step reasoning structure proposed in prior chain-of-thought-based VLA methods. At the same time, CoGHP takes a step further by extending it to a hierarchical policy learning framework for offline goal-conditioned RL. We have added further explanation of the relationship between chain-of-thought and CoGHP in `Section 1 of the revised paper`.

---

> ### Author Response · Authors · 2025-11-21
> **Response to Reviewer 7rmc (2/4)**
>
> > **W2: The claim that reverse-order subgoal generation is better is unsupported, and alternative formulations may be more principled.**
>
> Thank you for pointing out the lack of an ablation on subgoal generation order. Our initial design assumed that subgoals closer to the current state should aggregate more comprehensive information from the hierarchical reasoning process, and we therefore generated subgoals from the one farthest from the current state to the one closest to it. To test this assumption, we added an ablation that compares forward-order generation, which generates from the subgoal closest to the current state to the farthest subgoal, with reverse-order generation. We conducted experiments on navigation tasks where generating multiple subgoals yields stable performance, and examined how environment difficulty and the number of generated subgoals $H$ affect the impact of the subgoal generation order (`Appendix C.5`). In the easier antmaze-large environment, forward and reverse generation perform similarly. In contrast, in antmaze-giant, reverse generation outperforms forward generation, and the performance gap widens as $H$ increases. These findings support the validity of our initial design assumption.
>
> |  | forward (H=2) | forward (H=5) | reverse (H=2) | reverse (H=5) |
> | --- | --- | --- | --- | --- |
> | antmaze-large-navigate-v0: | 90$\pm$2 | 92$\pm$2 | 90$\pm$3 | 92$\pm$2 |
> | antmaze-giant-navigate-v0: | 71$\pm$2 | 50$\pm$5 | 78$\pm$8 | 61$\pm$7 |
>
> ---
>
> In response to the comment that the subgoal sequence should be optimized jointly, we note that CoGHP already satisfies this requirement. In CoGHP, a single network maps the state and goal to a sequence consisting of latent subgoals followed by the final action, and a shared goal-conditioned value function defines a loss over this entire sequence (`Eq. 9`). Because the loss is defined on the full sequence, all subgoals and the primitive action are optimized jointly within one network rather than being learned in separate stages.
>
> ---
>
> We appreciate the reviewer’s insightful suggestions regarding alternative formulations. As noted by the reviewer, bidirectional generation and diffusion-style iterative refinement that more directly optimize the subgoal sequence are natural extensions. For instance, one could use a bidirectional model to generate the full subgoal sequence with bidirectional context, or introduce a diffusion planner in CoGHP’s subgoal space to perform coarse-to-fine refinement. Such extensions, however, require engineering efforts that go beyond the scope of the current paper, so here we focus on how effective the simplest causal autoregressive structure can be on long-horizon offline goal-conditioned RL problems. In the `Section D of the revised paper`, we stated explicitly that (i) the current work considers only unidirectional subgoal generation, and (ii) extending the CoGHP framework with bidirectional generation and diffusion-based iterative refinement suggested by the reviewer is an important direction for future work.

---

> ### Author Response · Authors · 2025-11-21
> **Response to Reviewer 7rmc (3/4)**
>
> > **W3: The claimed generality of handling other subgoal types is non-trivial and requires further clarification.**
>
> Thank you for raising the question of how the framework could handle other forms of subgoal representation. Structurally, the MLP-Mixer backbone in CoGHP can also accommodate subgoals such as learned skill primitives [3] or abstract semantic embeddings [4], but, as the reviewer notes, doing so would require additional modalities or annotations in the dataset, a new definition of subgoal semantics, and matching training objectives. For example, using learned skill primitives as subgoals would entail first learning skill embeddings and a decoder from offline play data, then defining CoGHP’s subgoal tokens as these skill latents, and finally training the shared value function over the state–goal–skill space. Likewise, using language-based semantic subgoals would require language annotations for each subgoal, a pretrained language encoder, and an additional objective to align language embeddings with the future state distribution. `Section 4.4 of the revised version` explicitly states that the current implementation of CoGHP is restricted to subgoals based on encoded future states and that extending the framework to other subgoal types is left as future work, and the `Section D` has been updated to describe the additional modifications or assumptions needed to support such extensions.
>
> Although the current implementation of CoGHP is restricted to subgoals based on encoded future states, we also examined how this representation behaves under more complex observations. In particular, we visualized subgoals in the pixel-based visual-antmaze environment and included these results in the `Appendix C.10`. These examples show that, even under pixel-based observations, the agent still forms an intermediate subgoal that connects the current state to the goal, indicating that CoGHP’s subgoal representation extends stably to more complex observations.
>
> > **Q1: Why does the MLP-Mixer perform better?**
>
> By “stepwise procedural consistency,” we intended to highlight that each token in CoGHP has a fixed role determined by its position, such as state, goal, latent subgoal sequence, and action. In `Section 4.1 of the revised version`, we updated the wording to the clearer term “position-dependent token roles” and explained this meaning explicitly.
>
> We interpret the performance difference between the Transformer baseline and CoGHP as largely stemming from how each architecture handles position-dependent tokens. In CoGHP, unlike text in LLMs where tokens have context-dependent meanings and roles, the input sequence is composed of structured position-dependent token roles, where each index has a fixed semantic function as “current state, final goal, sequential intermediate subgoals and primitive action.” In such settings, prior time-series studies [5, 6] have observed that when the underlying signal is governed mainly by fixed position-dependent structure rather than rich context-dependent interactions across covariates, multivariate Transformer models can suffer from overfitting and degraded generalization, whereas time-step-dependent linear or MLP-based models tend to remain more robust. These results suggest that when token roles are relatively fixed and the signal is primarily position-dependent, the additional flexibility of data-dependent self-attention does not necessarily yield better generalization, making an MLP-Mixer backbone a natural architectural choice. Since the token roles are clearly fixed in CoGHP, this structural property helps explain the empirical Mixer-Transformer performance gap reported in the main text.

---

> ### Author Response · Authors · 2025-11-21
> **Response to Reviewer 7rmc (4/4)**
>
> > **Q2: Does the method reduce to HIQL when $H=1$, and how does performance vary with the number of predicted subgoals?**
>
> We appreciate the reviewer’s question regarding the structural relationship between CoGHP and HIQL. We clarify that even when the planning horizon $H$ is 1, CoGHP does not reduce to HIQL. In HIQL, the high-level and low-level policies are separate networks trained independently. In contrast, CoGHP uses a single MLP-Mixer backbone and a shared goal-conditioned value function to jointly train the entire hierarchy end-to-end, even when $H = 1$. Thus, reducing the number of subgoals to one does not collapse the framework into HIQL, and the unified, jointly trained structure remains intact.
>
> As noted in `Appendix C.2`, we report an ablation on the number of predicted subgoals, including a zero-subgoal setting. Navigation and manipulation tasks show different sensitivities to $H$. In navigation, performance clearly improves when subgoals are present compared to the zero-subgoal case. Although performance varies with the number of subgoals, the magnitude of the difference is not very large. In contrast, in manipulation tasks, too many subgoals can harm performance, and the zero-subgoal configuration can even outperform settings with multiple subgoals. This difference arises because, in navigation, latent subgoals mainly act as coarse waypoints that indicate intermediate directions or positions, whereas in manipulation, which requires more complex control, an overly fine-grained subgoal chain may impose excessive constraints on the low-level policy and hinder precise control. As mentioned in the limitation section, designing methods that are robust to the number of subgoals or that automatically determine the optimal number of subgoals for each task is an interesting direction for future work.
>
>
> [1] Zawalski, Michał, et al. "Robotic control via embodied chain-of-thought reasoning." *arXiv preprint arXiv:2407.08693* (2024).
> [2] Zhao, Qingqing, et al. "Cot-vla: Visual chain-of-thought reasoning for vision-language-action models." *Proceedings of the Computer Vision and Pattern Recognition Conference*. 2025.
> [3] Pertsch, Karl, Youngwoon Lee, and Joseph Lim. "Accelerating reinforcement learning with learned skill priors." Conference on robot learning. PMLR, 2021.
> [4] Ahn, Michael, et al. "Do as i can, not as i say: Grounding language in robotic affordances." arXiv preprint arXiv:2204.01691 (2022).
> [5] Zeng, Ailing, et al. "Are transformers effective for time series forecasting?." *Proceedings of the AAAI conference on artificial intelligence*. Vol. 37. No. 9. 2023.
> [6] Chen, Si-An, et al. "TSMixer: An All-MLP Architecture for Time Series Forecast-ing." *Transactions on Machine Learning Research*.

---

### Official Review · Reviewer_DvEG · 2025-10-30

**Soundness:** 2
**Presentation:** 3
**Contribution:** 2
**Rating:** 4
**Confidence:** 3

**Summary:**

This paper introduces Chain-of-Goals Hierarchical Policy (CoGHP), a method for offline goal-conditioned reinforcement learning that the authors mention draws inspiration from chain-of-thought reasoning in large language models to generate sequences of latent subgoals and actions using a unified MLP-Mixer architecture. The authors claim that the approach addresses limitations in traditional two-level hierarchical RL by enabling end-to-end training and multiple subgoal generation in reverse order (farthest to nearest). The main contributions include adapting MLP-Mixer for RL sequence modeling with a causal mixer, using advantage-weighted regression with a shared value function, and demonstrating performance gains on OGBench navigation (pointmaze, antmaze) and manipulation (cube, scene) tasks.

**Strengths:**

CoGHP demonstrates solid empirical performance on long-horizon tasks vs HIQL, highlighting benefits of multi-subgoal generation for navigation. The unified MLP-Mixer architecture enables end-to-end training, reducing the fragmentation in separate network methods like HIQL, and ablations confirm the causal mixer's role in complex settings.

**Weaknesses:**

1. It seems like the chain-of-thought inspiration is superficial rather than intuitive or practical: latent subgoals are opaque embeddings, not explicit reasoning steps, making the LLM analogy more rhetorical than substantive. The LLM analogy is conceptually motivating but technically superficial. The real contributions are (a) unified autoregressive generation and (b) end-to-end training. Also, there is no ablation to test forward subgoal generation, leaving the reverse-order claim unverified.

2. The baselines are limited to OGBench standards, which might question robustness to other domains.

3. The limitations like distribution shift vulnerability are acknowledged but untested, with no OOD experiments despite offline RL's emphasis on generalization.

4. Computational overhead (training/inference time) is unreported, hindering practical assessment against baselines.

**Questions:**

1. Why generate subgoals from farthest to nearest? Can you provide an ablation comparing this to nearest-to-farthest ordering on antmaze-giant and cube-triple to justify the design.

2. How does CoGHP handle OOD goals? Can you provide evaluations on success rates on goals outside the dataset distribution, as offline RL often faces unseen targets.

3. Please provide comparisons of runtime and parameters to HIQL: Report training time, inference latency, and FLOPs for fair efficiency claims.​

4. Does CoGHP generalize to diverse robotic manipulation tasks beyond the curated OGBench setup? Like MetaWorld Benchmark (Robotics).

---

> ### Author Response · Authors · 2025-11-21
> **Response to Reviewer DvEG (1/3)**
>
> We sincerely thank you for your valuable feedback on our work. Below are our responses to the feedback provided by the reviewer DvEG.
>
> > **W1 & Q1: The CoT analogy appears superficial, and the justification for reverse-order subgoal generation lacks ablation support.**
>
> Thank you for pointing out the relationship between CoGHP and chain-of-thought. As the reviewer notes, the subgoals used in CoGHP are latent embeddings rather than explicit representations such as environment states or natural language rationales, so each subgoal does not immediately appear as a human-interpretable reasoning step. However, the chain-of-thought perspective in this work instead emphasizes “thinking step-by-step.” More specifically, we focused on (i) unfolding a complex problem into a sequence of intermediate tokens, (ii) keeping this sequence conditional on the original query while passing through intermediate steps to produce the final output, and (iii) handling the entire process within a single unified sequence model. CoGHP builds on this perspective to mitigate the limitations of prior offline hierarchical RL methods that rely on two separately trained networks. Specifically, it (i) unfolds goals that are difficult to solve with a direct state–action mapping into a sequence of latent subgoal tokens, (ii) keeps this sequence conditional on the final goal while generating primitive actions through these intermediate subgoals, and (iii) processes the whole hierarchy end-to-end within a single unified model. This chain-of-thought-style “input - intermediate reasoning steps - primitive action” structure is consistent with recent works on robotic control with vision-language-action models (VLA), such as ECoT [1] and CoT-VLA [2], in which the model first produces intermediate embodied or visual reasoning steps and then predicts actions. In this sense, our work inherits the step-by-step reasoning structure proposed in prior chain-of-thought-based VLA methods. At the same time, CoGHP takes a step further by extending it to a hierarchical policy learning framework for offline goal-conditioned RL. Therefore, the unified autoregressive generation and end-to-end training highlighted by the reviewer correspond to the concrete architectural design that implements chain-of-thought-style multistep intermediate reasoning within a single network. We have added further explanation of the relationship between chain-of-thought and CoGHP in `Section 1 of the revised paper`. Additionally, although inference uses only latent subgoal embeddings, jointly training an auxiliary decoder allows the generated subgoals to be reconstructed into coordinate or visual space, as shown in `Section 5.4` and the `Appendix C.10`, providing partial interpretation and visualization of the learned subgoal sequence.
>
> ---
>
> Thank you for pointing out the lack of an ablation on subgoal generation order. Our initial design assumed that subgoals closer to the current state should aggregate more comprehensive information from the hierarchical reasoning process, and we therefore generated subgoals from the one farthest from the current state to the one closest to it. To test this assumption, we added an ablation that compares forward-order generation, which generates from the subgoal closest to the current state to the farthest subgoal, with reverse-order generation. We conducted experiments on navigation tasks where generating multiple subgoals yields stable performance, and examined how environment difficulty and the number of generated subgoals $H$ affect the impact of the subgoal generation order (`Appendix C.5`). In the easier antmaze-large environment, forward and reverse generation perform similarly. In contrast, in antmaze-giant, reverse generation outperforms forward generation, and the performance gap widens as $H$ increases. These findings support the validity of our initial design assumption.
>
> |  | forward (H=2) | forward (H=5) | reverse (H=2) | reverse (H=5) |
> | --- | --- | --- | --- | --- |
> | antmaze-large-navigate-v0: | 90$\pm$2 | 92$\pm$2 | 90$\pm$3 | 92$\pm$2 |
> | antmaze-giant-navigate-v0: | 71$\pm$2 | 50$\pm$5 | 78$\pm$8 | 61$\pm$7 |

---

> ### Author Response · Authors · 2025-11-21
> **Response to Reviewer DvEG (2/3)**
>
> > **W2 & Q4: The experimental scope is limited to OGBench, raising concerns about cross-domain robustness.**
>
> Thank you for asking whether experiments confined to OGBench are sufficient to demonstrate generalization to other robotic manipulation domains. OGBench is a recently proposed benchmark for offline goal-conditioned RL that is already widely used in the literature. Because it is designed to directly test stitching, long-horizon reasoning, and multi-goal generalization across both navigation and manipulation tasks, we considered it a particularly appropriate benchmark for evaluating CoGHP. Within OGBench, CoGHP is evaluated on point and ant maze navigation, robotic manipulation tasks such as Cube and Scene, and pixel-based tasks including visual-antmaze and visual-cube. These environments differ substantially in observation dimensionality, action spaces, and the behavior patterns required from the agent. Nevertheless, CoGHP achieves consistently higher performance than prior offline goal-conditioned RL algorithms in most of these domains. Because OGBench covers heterogeneous navigation, manipulation, and visual control tasks, consistently strong performance across these environments suggests that CoGHP is not restricted to a single narrow domain. In this sense, evaluation on OGBench already provides a reasonably broad indication of its generalization ability.
>
> MetaWorld is widely used as an online benchmark for multi-task and meta-RL. In contrast, the focus of CoGHP is to propose a new offline goal-conditioned RL method that efficiently solves long-horizon goal-reaching tasks, so we considered OGBench to be a suitable benchmark for this study. In the future, if off-policy manipulation datasets of comparable quality to OGBench become available for MetaWorld, extending CoGHP to that benchmark should be feasible.
>
> > **W3 & Q2: The framework lacks OOD evaluations, leaving unseen-goal generalization untested.**
>
> As the reviewer points out, performance in out-of-distribution (OOD) settings is still an important issue in offline RL. However, the main focus of this work was on addressing challenges of offline goal-conditioned RL, such as stitching, long-horizon reasoning, and multi-goal generalization. OGBench was also designed to target these aspects rather than setups for OOD environments. The limitation section explicitly states that the performance of CoGHP may degrade when moving to environments or tasks that are substantially different from the training data. Nevertheless, CoGHP can handle OOD goals to some extent. In the OGBench Cube environment, the offline dataset is collected by a random pick-and-place policy, so specific evaluation goal configurations may be rarely, if ever, observed in the data. As described in the OGBench paper, this setting can therefore be viewed as requiring some degree of unseen goal generalization. In our experiments, CoGHP achieves consistently higher success rates than previous offline GCRL algorithms on several variants of Cube (cube-single, cube-double, cube-triple), which suggests that, at least within the same environment, CoGHP has a certain level of unseen goal generalization ability for goal configurations that appear only rarely in the data.

---

> ### Author Response · Authors · 2025-11-21
> **Response to Reviewer DvEG (3/3)**
>
> > **W4 & Q3: Provide computational overhead and runtime comparisons.**
>
> On a single NVIDIA GeForce RTX 3090 GPU, using the antmaze-giant environment as reference, the training time, number of parameters, inference latency, and FLOPs of HIQL and CoGHP are as follows.
>
> | Method | Training Time | #Params | Inference Latency | FLOPs |
> | --- | --- | --- | --- | --- |
> | HIQL | 3 h | 3.8 M | 0.8 ms | 2.9 M |
> | CoGHP | 5 h | 5.5 M | 4.4 ms | 7.1 M |
>
> As can be seen from the table, CoGHP uses more parameters and computation than HIQL, and its training time and inference latency are also higher on antmaze-giant. However, an inference time of about 4.4 ms is unlikely to be a significant limitation for real-time applications in robot control and simulation environments.
>
> To directly test whether model size explains our performance gains, we scaled up HIQL to match CoGHP’s parameter count and reran experiments on antmaze-giant and cube-double. Increasing HIQL’s capacity did not yield any significant improvement. Its performance remained well below that of CoGHP across all tested environments. This suggests that our observed gains are primarily attributable to the unified autoregressive architecture, rather than additional parameters alone.
>
> | Env | HIQL | HIQL (Large parameter) | CoGHP (Ours) |
> | --- | --- | --- | --- |
> | antmaze-giant-navigate-v0 | 65$\pm$5 | 67$\pm$5 | 78$\pm$8 |
> | cube-double-noisy-v0 | 2$\pm$1 | 2$\pm$1 | 54$\pm$5 |
>
>
> [1] Zawalski, Michał, et al. "Robotic control via embodied chain-of-thought reasoning." *arXiv preprint arXiv:2407.08693* (2024).
> [2] Zhao, Qingqing, et al. "Cot-vla: Visual chain-of-thought reasoning for vision-language-action models." *Proceedings of the Computer Vision and Pattern Recognition Conference*. 2025.

---

### Official Review · Reviewer_chFA · 2025-10-31

**Soundness:** 2
**Presentation:** 2
**Contribution:** 2
**Rating:** 4
**Confidence:** 3

**Summary:**

The paper proposes Chain-of-Goals Hierarchical Policy, a unified policy that generates a short sequence of latent subgoals followed by the final action in a single forward pass. The key idea is to treat subgoals as reasoning tokens, produced autoregressively so that later predictions condition on earlier ones while remaining aware of the original goal. The architecture uses an MLP-Mixer backbone with a learnable causal token mixer, trained end to end with a shared goal-conditioned value function and advantage-weighted objectives. Experiments on OGBench navigation and manipulation tasks report sizable gains over strong offline goal-conditioned RL baselines, with ablations suggesting benefits from the Mixer backbone and the causal mixer.

**Strengths:**

* The paper targets a pain point in offline goal-conditioned RL for long horizons and argues for a cohesive alternative to two-level hierarchies. Formulating hierarchical control as autoregressive subgoal generation inside one network is conceptually neat and practically appealing, since it preserves access to the final goal and allows gradients to flow through all stages.

* Empirically, the method shows strong results across diverse domains. Notably, it improves success on difficult giant mazes and complex manipulation sequences where multiple intermediate decisions matter. These gains are consistent with the claim that multi-step intermediate guidance helps long-horizon tasks.

* The ablations are helpful. Replacing the Mixer with a Transformer hurts on the hardest tasks, and removing the causal mixer degrades performance further, which supports the specific architectural choices rather than attributing wins to generic capacity.

**Weaknesses:**

* The novelty is partly architectural refactoring and framing. The chain-of-thought analogy is evocative, but the subgoals are supervised by fixed k-step future states from trajectories. This is closer to structured imitation with value-based weighting than to learned reasoning.
The methodology introduces potential train–test mismatch. Training uses teacher forcing with ground-truth subgoals, while inference relies on the model’s own subgoal predictions. The paper acknowledges teacher forcing but does not quantify error accumulation or compare against scheduled sampling or consistency regularizers designed for autoregressive rollouts.

* Some comparisons and analyses feel underdeveloped. The claim that Transformers are less suitable is asserted and supported by a single ablation, but hyperparameter parity and capacity normalization are not deeply probed. It is also unclear how sensitive results are to the choice of k for subgoal extraction, to the advantage temperature, and to the exact weighting between subgoal and action losses. The experiments report strong headline numbers, yet each environment evaluates only five predefined state–goal pairs and success rates are averaged across eight seeds, which makes the statistical picture somewhat narrow. More per-task breakdowns or success-vs-horizon plots would improve confidence.

* On the representation side, most subgoals are encoded future states. The paper briefly notes that other subgoal types would fit, but the only concrete visual evidence comes from decoding to coordinates for antmaze. Demonstrations of learned abstract subgoals or skills, or at least richer visualizations across tasks, would better support the generality claim.

**Questions:**

* How robust is performance to the choice of H and the spacing k used to sample supervision subgoals from trajectories? A sensitivity sweep that varies H and k together would help establish whether improvements persist beyond the selected settings.

* Can the authors quantify exposure bias from teacher forcing? For example, report success when the policy is rolled out with its own predicted subgoals but trained without teacher forcing, or include scheduled sampling. A plot of success vs rollout depth of predicted subgoals would clarify compounding error.

* What is the effect of the causal mixer relative to simpler positional encodings or strictly triangular masking without learnable mixing? An ablation that replaces the causal mixer with fixed lower-triangular averaging could isolate the benefit of learnability.

* How fair and capacity-matched are the Transformer baselines? Please provide layer counts, parameter totals, token dimensions, and training curves for Mixer vs Transformer across tasks to rule out under-tuning.

* Do results hold when evaluating on larger sets of randomly sampled state–goal pairs and under distribution shift in goals? Reporting confidence intervals across many goals, as well as success vs geodesic distance to goal, would strengthen the empirical case.

* Can the method operate with non-state subgoals, such as latent skills or semantic waypoints, and still retain advantages? A small-scale experiment with learned discrete subgoals would substantiate the generality claim.

---

> ### Author Response · Authors · 2025-11-21
> **Response to Reviewer chFA (1/4)**
>
> We sincerely thank you for your valuable feedback on our work. Below are our responses to the feedback provided by the reviewer chFA.
>
> > **W1 & Q2: The supervision resembles structured imitation rather than learned reasoning, and teacher-forcing ablations are needed.**
>
> Thank you for pointing out the relationship between CoGHP and chain-of-thought. We regard the core of chain-of-thought as “thinking step-by-step”: (i) unfolding a complex problem into a sequence of intermediate tokens, (ii) keeping this sequence conditional on the original query while passing through intermediate steps to produce the final output, and (iii) handling the entire process within a single unified sequence model. CoGHP builds on this perspective to mitigate the limitations of prior offline hierarchical RL methods that rely on two separately trained networks. Specifically, it (i) unfolds goals that are difficult to solve with a direct state–action mapping into a sequence of latent subgoal tokens, (ii) keeps this sequence conditional on the final goal while generating primitive actions through these intermediate subgoals, and (iii) processes the whole hierarchy end-to-end within a single unified model. This chain-of-thought-style “input - intermediate reasoning steps - primitive action” structure is consistent with recent works on robotic control with vision-language-action models (VLA), such as ECoT [1] and CoT-VLA [2], in which the model first produces intermediate embodied or visual reasoning steps and then predicts actions. In this sense, our work inherits the step-by-step reasoning structure proposed in prior chain-of-thought-based VLA methods. At the same time, CoGHP takes a step further by extending it to a hierarchical policy learning framework for offline goal-conditioned RL. We have added further explanation of the relationship between chain-of-thought and CoGHP in `Section 1 of the revised paper`.
>
> In addition, as the reviewer notes, CoGHP uses fixed k-step future states from demonstrations and advantage-weighted regression (AWR) in training, so it can reasonably be viewed as structured imitation with value-based weighting. However, since it relies on a value function trained with the offline RL algorithm Implicit Q-Learning (IQL), we view the formulation as closer to an extension of offline RL than to imitation learning.
>
> ---
>
> Teacher forcing is a standard training scheme for models that autoregressively generate intermediate reasoning tokens and actions. CoGHP adopts this training method. As the reviewer notes, conditioning on ground-truth subgoals during training and on model-predicted subgoals at test time can introduce a train-test mismatch. In our experiments, however, CoGHP generates only a small number of reasoning tokens, so additional mechanisms such as scheduled sampling or consistency regularizers were not necessary. Instead, to quantify the effect of teacher forcing as suggested in Q2, we ran an ablation that compares training with teacher forcing and training without teacher forcing under the same setting. The results of this experiment are reported in the `Appendix C.8`. When the policy is trained without teacher forcing and rolled out based on its own predicted subgoals, the success rate clearly decreases, indicating that standard teacher forcing is important for stable training and robust long-horizon rollouts in the proposed CoGHP architecture.
>
> |  | w/o teacher forcing | Ours |
> | --- | --- | --- |
> | antmaze-giant-navigate-v0: | 18$\pm$5 | 78$\pm$8 |
> | cube-double-noisy-v0 | 3$\pm$2 | 54$\pm$5 |

---

> ### Author Response · Authors · 2025-11-21
> **Response to Reviewer chFA (2/4)**
>
> > **W2 & Q1: Broader ablation analyses is needed.**
>
> As requested, we performed a sensitivity sweep that varies subgoal steps $k$ and number of subgoals $H$ together and additional ablations on the advantage temperature, and we added these results to the `Appendix C.3 and C.9`. We also report the ablation on the weighting coefficient $\lambda_h$ for the subgoal loss in the `Appendix C.7`. In the joint $H$–$k$ sweep (`Appendix C.3`), navigation tasks are more sensitive to the spacing $k$ between subgoals than to the number of subgoals $H$, whereas manipulation tasks are more sensitive to $H$. We interpret these differences as arising from task-specific characteristics. In navigation tasks, subgoals mainly serve as coarse waypoints that indicate intermediate directions or positions, so as long as they are spaced reasonably, performance is not highly sensitive to the exact number of subgoals. By contrast, in manipulation tasks, which require more precise control, an overly fine-grained subgoal chain can overconstrain the low-level policy and hinder accuracy. When $H = 1$ and the subgoal spacing is kept around {5, 10, 20}, however, performance is relatively less sensitive to $k$. This indicates that subgoal design interacts with task characteristics and supports the point made in the limitation section that developing methods which are robust to the number of subgoals, or can automatically select an optimal subgoal horizon for each task, is an important direction for future work.
>
> For the advantage temperature, comparing the values {1, 3, 10} shows that 3 achieves the best performance, and we do not observe a sharp performance collapse at 1 or 10, suggesting that the proposed method is reasonably robust to this hyperparameter. For the weighting coefficient, we fixed $\lambda_\ell = 1$ and ran experiments with different values of $\lambda_h$. With a single predicted sub-goal ($H=1$, e.g., cube-double), performance is stable over a wide span of $\lambda_h$, but when two sub-goals are generated ($H=2$, e.g., antmaze-giant), setting $\lambda_h$ too high rapidly destabilises training and drives success toward zero. We found that using the heuristic value $1/k$ yields overall stable behavior ($\lambda_h=1/50$ for antmaze-giant).
>
> | advantage temperature | 1 | 3 | 10 |
> | --- | --- | --- | --- |
> | antmaze-giant-navigate-v0 | 75$\pm$2 | 78$\pm$8 | 61$\pm$6 |
> | cube-double-noisy-v0 | 47$\pm$2 | 54$\pm$5 | 50$\pm$4 |
>
> | weighting coefficient $\lambda_h$ | 10 | 1 | 0.1 | 0.02 | 0.01 |
> | --- | --- | --- | --- | --- | --- |
> | antmaze-giant-v0-navigate | 0$\pm$0 | 2$\pm$1 | 66$\pm$4 | 79$\pm$8 | 65$\pm$4 |
> | cube-double-noisy-v0 | 52$\pm$4 | 51$\pm$1 | 54$\pm$5 | 55$\pm$9 | 56$\pm$7 |
>
> > **W2 & Q4: How fair and capacity-matched are the Transformer baselines?**
>
> Thank you for raising the question of fairness and capacity matching for the Transformer baseline. In response, `Appendix B.3 and C.4` now reports the number of layers, parameter counts, token dimensions, and training curves for both Mixer and Transformer. We configured these experiments so that the Mixer–Transformer comparison in this work is fair and capacity-matched with respect to both model size and training configuration. The Transformer baseline uses 2 layers, and the token dimension is matched to CoGHP’s state embedding dimensions in all environments. The parameter counts are also closely aligned, for example, 5.61M for the Transformer and 5.54M for CoGHP on antmaze-giant. Both models are trained with the same optimizer, learning rate schedule, and number of training steps. The training curves in the `Figure 7` show that the Transformer converges stably, with no indication of under-tuning.
>
> We interpret the performance difference between the Transformer baseline and CoGHP as largely stemming from how each architecture handles position-dependent tokens. In CoGHP, unlike text in LLMs where tokens have context-dependent meanings and roles, the input sequence is composed of structured position-dependent token roles, where each index has a fixed semantic function as “current state, final goal, sequential intermediate subgoals and primitive action.” In such settings, prior time-series studies [5, 6] have observed that when the underlying signal is governed mainly by fixed position-dependent structure rather than rich context-dependent interactions across covariates, multivariate Transformer models can suffer from overfitting and degraded generalization, whereas time-step-dependent linear or MLP-based models tend to remain more robust. These results suggest that when token roles are relatively fixed and the signal is primarily position-dependent, the additional flexibility of data-dependent self-attention does not necessarily yield better generalization, making an MLP-Mixer backbone a natural architectural choice. Since the token roles are clearly fixed in CoGHP, this structural property helps explain the empirical Mixer-Transformer performance gap.

---

> ### Author Response · Authors · 2025-11-21
> **Response to Reviewer chFA (3/4)**
>
> > **W2 & Q5: More per-task breakdowns would improve confidence.**
>
> Thank you for raising concerns about statistical reliability and generalization over the goal distribution. Our experiments follow the standard OGBench evaluation protocol, in which each task specifies five predefined state–goal pairs and performance is measured as the average success rate over these goals across multiple seeds. Because OGBench is designed to test key challenges in offline goal-conditioned RL, including stitching, long-horizon reasoning, and multi-goal generalization, we regard this setup as an appropriate basis for assessing the benefits of CoGHP.
>
> As the reviewer suggested, we added per-task breakdowns to the `Appendix C.11` for a more detailed analysis. The OGBench paper does not characterize difficulty differences across the five evaluation settings, but in manipulation domains such as cube and scene we observe a tendency for higher-index tasks (closer to task 5) to require more complex behavior. For each environment, we report success rates for all five evaluation goals. Although performance varies with goal difficulty within the same environment, CoGHP outperforms the baseline algorithms on most tasks, indicating that its improvements are consistent not only in headline averages but also at the level of individual goals.
>
> ---
>
> Regarding the distribution shift in goals, CoGHP can handle OOD goals to some extent. In the OGBench Cube environment, the offline dataset is collected by a random pick-and-place policy, so specific evaluation goal configurations may be rarely, if ever, observed in the data. As described in the OGBench paper, this setting can therefore be viewed as requiring some degree of unseen goal generalization. In our experiments, CoGHP achieves consistently higher success rates than previous offline GCRL algorithms on several variants of Cube (cube-single, cube-double, cube-triple), which suggests that, at least within the same environment, CoGHP has a certain level of unseen goal generalization ability for goal configurations that appear only rarely in the data.
>
> > **Q3: Clarify the benefit of the causal mixer through a ablation.**
>
> Thank you for pointing out the need for additional ablations on the causal mixer. Regarding positional encoding, the MLP-Mixer backbone in CoGHP uses a token-mixing structure that applies an MLP over the entire token dimension and treats each position as a fixed index. Because positional information is already encoded structurally in this way, we do not introduce any separate positional encoding, in contrast to Transformers. The role of the causal mixer is to provide a causal inductive bias when CoGHP takes the state and goal as input and autoregressively generates the latent subgoal sequence and final action in sequence, so that each token focuses more on past reasoning tokens. This design is analogous to causal attention in LLMs, in that the current token aggregates information primarily from past subgoal tokens rather than future tokens. To allow the model to learn how to perform this mixing, we made the causal weights learnable.
>
> As suggested by the reviewer, to isolate the effect of the learnable causal mixer, we ran an ablation that compares (i) a variant that completely removes the causal mixer, (ii) a non-learnable causal mixer that replaces the learnable weights with fixed lower-triangular averaging, and (iii) default CoGHP with a learnable causal mixer. The results of this experiment are reported in the `Appendix C.6`. In most environments, (i) removing the causal mixer and (ii) fixed lower-triangular averaging yield similar performance to each other, and both consistently underperform (iii) with the learnable causal mixer. This gap becomes more pronounced as task complexity increases. Thus, this experiment shows that simple causal masking or fixed averaging is not sufficient. It also indicates that a learnable causal mixer that learns the weights over past reasoning tokens plays a meaningful role in improving performance, especially on complex long-horizon tasks.
>
> |  | w/o causal-mixer | fixed causal-mixer | CoGHP (ours) |
> | --- | --- | --- | --- |
> | antmaze-medium-navigate-v0 | 97$\pm$1 | 97$\pm$1 | 97$\pm$2 |
> | antmaze-giant-navigate-v0 | 71$\pm$7 | 72$\pm$1 | 78$\pm$8 |
> | cube-single-noisy-v0 | 95$\pm$4 | 98$\pm$2 | 97$\pm$3 |
> | cube-double-noisy-v0 | 44$\pm$4 | 51$\pm$8 | 54$\pm$5 |
> | cube-triple-noisy-v0 | 27$\pm$6 | 27$\pm$4 | 42$\pm$3 |

---

> ### Author Response · Authors · 2025-11-21
> **Response to Reviewer chFA (4/4)**
>
> > **W3 & Q6: Evidence for handling abstract or non-state subgoals is insufficient.**
>
> Thank you for raising the question of how the framework could handle other forms of subgoal representation. Structurally, the MLP-Mixer backbone in CoGHP can also accommodate subgoals such as learned skill primitives [5] or abstract semantic embeddings [6], but doing so would require additional modalities or annotations in the dataset, a new definition of subgoal semantics, and matching training objectives.
>
> For example, using learned skill primitives as subgoals would entail first learning skill embeddings and a decoder from offline play data, then defining CoGHP’s subgoal tokens as these skill latents, and finally training the shared value function over the state–goal–skill space. Likewise, using language-based semantic subgoals would require language annotations for each subgoal, a pretrained language encoder, and an additional objective to align language embeddings with the future state distribution. `Section 4.4 in the revised paper` explicitly states that the current implementation of CoGHP is restricted to subgoals based on encoded future states and that extending the framework to other subgoal types is left as future work, and the `Section D` has been updated to describe the additional modifications or assumptions needed to support such extensions.
>
> We agree that the visual evidence in the main text is overly focused on decoding latent subgoals to coordinates in antmaze. In response, we have added additional visualizations of the latent subgoals generated by CoGHP in the visual antmaze environment to the `Appendix C.10`. These examples show that, under pixel-based observations the agent still forms a intermediate subgoal that connect the current state to the goal, supporting that CoGHP’s subgoal representation extends stably to more complex observations.
>
> [1] Zawalski, Michał, et al. "Robotic control via embodied chain-of-thought reasoning." *arXiv preprint arXiv:2407.08693* (2024).
> [2] Zhao, Qingqing, et al. "Cot-vla: Visual chain-of-thought reasoning for vision-language-action models." *Proceedings of the Computer Vision and Pattern Recognition Conference*. 2025.
> [3] Zeng, Ailing, et al. "Are transformers effective for time series forecasting?." *Proceedings of the AAAI conference on artificial intelligence*. Vol. 37. No. 9. 2023.
> [4] Chen, Si-An, et al. "TSMixer: An All-MLP Architecture for Time Series Forecast-ing." *Transactions on Machine Learning Research*.
> [5] Pertsch, Karl, Youngwoon Lee, and Joseph Lim. "Accelerating reinforcement learning with learned skill priors." Conference on robot learning. PMLR, 2021.
> [6] Ahn, Michael, et al. "Do as i can, not as i say: Grounding language in robotic affordances." arXiv preprint arXiv:2204.01691 (2022).

---

### Official Review · Reviewer_yZJW · 2025-11-21

**Soundness:** 3
**Presentation:** 3
**Contribution:** 2
**Rating:** 4
**Confidence:** 4

**Summary:**

This work introduces the framework Chain-of-Goals Hierarchical Policy (CoGHP). Inspired by chain-of-through reasoning in large language models, this framework formulates hierarchical control problems with autoregressive sequence. In a single forward pass, the framework generates a sequence of latent subgoals each encoding an intermediate decision-making, followed by the primitive action.
The work uses MLP-Mixer backbone with a learnable causal token mixer to capture consistent structural relationships. Experiments on long-horizon offline control tasks reflects non-trivial improvements over strong baselines. Ablation study shows that the benefits is from the mixer backbone.

**Strengths:**

* Originality: The idea of taking inspiration from Chain-of-Thought to reformulate HRL tasks with autoregressive sequence within a single unified network is novel. The work also shows that this formulation allows for end-to-end optimization, which is practical and efficient.

* Significance: The experiments show that an MLP-Mixer backbone consistently performs better than stronger baselines across various tasks that uses Transformer architecture. The experiment benchmark is also complex enough where it is often necessary to have multiple intermediate decisions. These results demonstrate that multi-step intermediate guidance helps long-horizon tasks.

* Quality and clarity: The ablations shows that the MLP-Mixer and the causal mixer is crucial, as replacing them decreases performance.

**Weaknesses:**

* Using the term chain-of-thought is a little misleading because while it is valid as an inspiration, in practice it doesn't seem to be directly influencing the architecture. The subgoals are not explicit interpretable reasonings like their counterparts in LLMs, rather they are embeddings supervised by fixed step future states. The work could perhaps make it more clear that their contribution is focused on proposing a unified autoregressive generation and highlight that this allows for end-to-end optimization, and discuss more on the relationship to the line of work that frame offline RL as sequence modeling problem.

* The ablation study feels under-explored: there could be more context discussing things like hyperparameter parity, how selecting different k affects the extracted subgoals. In general, this paper could benefit from more quantitative results that focuses on different tasks. The work could also benefit from adding an OOD experiments and discuss generalization as part of offline RL's focus. The claim that generating subgoals in reverse order also feels under-explored and lacks a clear and concrete supporting result.

**Questions:**

1. The reason behind generating subgoal in the order of farthest to nearest is not very clear. Is it really better compared to generating it the other way around, or is it just easier for the architecture?

---

> ### Author Response · Authors · 2025-11-21
> **Response to Reviewer yZJW (1/3)**
>
> We sincerely thank you for your valuable feedback on our work. Below are our responses to the feedback provided by the reviewer yZJW.
>
> > **W1-1: Usage of the term chain-of-thought is a little misleading. / The main contribution instead seems to be the unified autoregressive generation that enables end-to-end optimization.**
>
> Thank you for pointing out the relationship between CoGHP and chain-of-thought. We regard the core of chain-of-thought as “thinking step-by-step”: (i) unfolding a complex problem into a sequence of intermediate tokens, (ii) keeping this sequence conditional on the original query while passing through intermediate steps to produce the final output, and (iii) handling the entire process within a single unified sequence model. CoGHP builds on this perspective to mitigate the limitations of prior offline hierarchical RL methods that rely on two separately trained networks. Specifically, it (i) unfolds goals that are difficult to solve with a direct state–action mapping into a sequence of latent subgoal tokens, (ii) keeps this sequence conditional on the final goal while generating primitive actions through these intermediate subgoals, and (iii) processes the whole hierarchy end-to-end within a single unified model. This chain-of-thought-style “input - intermediate reasoning steps - primitive action” structure is consistent with recent works on robotic control with vision-language-action models (VLA), such as ECoT [1] and CoT-VLA [2], in which the model first produces intermediate embodied or visual reasoning steps and then predicts actions. In this sense, our work inherits the step-by-step reasoning structure proposed in prior chain-of-thought-based VLA methods. At the same time, CoGHP takes a step further by extending it to a hierarchical policy learning framework for offline goal-conditioned RL. Therefore, the unified autoregressive generation and end-to-end training highlighted by the reviewer correspond to the concrete architectural design that implements chain-of-thought-style multistep intermediate reasoning within a single network. We have added further explanation of the relationship between chain-of-thought and CoGHP in `Section 1 of the revised paper`.
>
> > **W1-2: The subgoals are not explicit interpretable reasonings.**
>
> As the reviewer notes, the subgoals used in CoGHP are latent embeddings rather than explicit representations such as environment states or natural language rationales, so each subgoal does not immediately appear as a human-interpretable reasoning step. Nonetheless, as discussed above, the chain-of-thought perspective in this work emphasizes “thinking step by step,” and human interpretability is therefore not a primary objective of this study. However, even though inference uses only latent subgoal embeddings, jointly training an auxiliary decoder allows the generated subgoals to be reconstructed into coordinate or visual space, as shown in `Section 5.4 and Appendix C.10`, providing partial interpretation and visualization of the learned subgoal sequence.
>
> > **W1-3: Discuss more on the relationship to the line of work that frames offline RL as a sequence modeling problem.**
>
> Thank you for connecting CoGHP to the “offline RL as sequence modeling” literature, such as Decision Transformer [3] and Trajectory Transformer [4]. We agree that CoGHP is related to these methods through its use of a sequence-based architecture. However, there is a difference in the role played by this sequence modeling. Decision Transformer and Trajectory Transformer encode offline trajectory as a single long sequence of (returns-to-go or rewards, s, a) and use it for return-conditioned action prediction (Decision Transformer) or beam-search-based planning (Trajectory Transformer). In contrast, CoGHP defines, at each decision step, a structured sequence of “current state, final goal, latent subgoal chain, and action” and performs hierarchical subgoal inference and action generation on this sequence. For this reason, while CoGHP shares the use of sequence models with Decision Transformer and Trajectory Transformer, our work emphasizes its chain-of-thought-style role of generating intermediate subgoal tokens and using them for hierarchical control, rather than encoding full trajectories.

---

> ### Author Response · Authors · 2025-11-21
> **Response to Reviewer yZJW (2/3)**
>
> > **W2-1: The ablation study feels under-explored: more discussion on hyperparameter parity and subgoal step $k$ are needed.**
>
> Thank you for asking whether the comparisons with other baselines were fair and capacity-matched. We address this for both the Transformer-based baseline and our main baseline, HIQL. For the Transformer-based baseline, We configured these experiments so that the Mixer–Transformer comparison in this work is fair and capacity-matched with respect to both model size and training configuration. The Transformer baseline uses 2 layers, and the token dimension is matched to CoGHP’s state embedding dimensions in all environments. The parameter counts are also closely aligned, for example, 5.61M for the Transformer and 5.54M for CoGHP on antmaze-giant. Both models are trained with the same optimizer, learning rate schedule, and number of training steps. The training curves in the `Figure 7` show that the Transformer converges stably, with no indication of under-tuning.
>
> For HIQL, on antmaze-giant the network has about 3.8M parameters, which is smaller than the 5.54M parameters used in CoGHP. To directly test whether model size explains our performance gains, we scaled up HIQL to match CoGHP’s parameter count and reran experiments on antmaze-giant and cube-double. Increasing HIQL’s capacity did not yield any significant improvement. Its performance remained well below that of CoGHP across all tested environments. This suggests that our observed gains are primarily attributable to the unified autoregressive architecture, rather than additional parameters alone.
>
> | Env | HIQL | HIQL (Large parameter) | CoGHP (Ours) |
> | --- | --- | --- | --- |
> | antmaze-giant-navigate-v0 | 65$\pm$5 | 67$\pm$5 | 78$\pm$8 |
> | cube-double-noisy-v0 | 2$\pm$1 | 2$\pm$1 | 54$\pm$5 |
>
> ---
>
> To assess the influence of the subgoal step $k$, we conducted experiments to examine how performance changes with $k$ across different environments and different numbers of generated subgoals $H$. The results can be found in `Appendix C.7`. In this experiment, navigation tasks are more sensitive to the spacing $k$ between subgoals than to the number of subgoals $H$, whereas manipulation tasks are more sensitive to $H$. We interpret these differences as arising from task-specific characteristics. In navigation tasks, subgoals mainly serve as coarse waypoints that indicate intermediate directions or positions, so as long as they are spaced reasonably, performance is not highly sensitive to the exact number of subgoals. By contrast, in manipulation tasks, which require more precise control, an overly fine-grained subgoal chain can overconstrain the low-level policy and hinder accuracy. When $H = 1$ and the subgoal spacing is kept around {5, 10, 20}, however, performance is relatively less sensitive to $k$.
>
> > **W2-2: More quantitative results that focuses on different tasks could be beneficial.**
>
> Thank you for raising the question about the breadth of tasks considered in our experiments. OGBench is a recently proposed benchmark for offline goal-conditioned RL that is already widely used in the literature. Because it is designed to directly test stitching, long-horizon reasoning, and multi-goal generalization across both navigation and manipulation tasks, we considered it a particularly appropriate benchmark for evaluating CoGHP. Within OGBench, CoGHP is evaluated on point and ant maze navigation, robotic manipulation tasks such as Cube and Scene, and pixel-based tasks including visual-antmaze and visual-cube. These environments differ substantially in observation dimensionality, action spaces, and the behavior patterns required from the agent. Nevertheless, CoGHP achieves consistently higher performance than prior offline goal-conditioned RL algorithms in most of these domains. Taken together, these results provide substantial evidence that CoGHP can effectively handle a diverse range of goal-conditioned tasks within this evaluation setting.

---

> ### Author Response · Authors · 2025-11-21
> **Response to Reviewer yZJW (3/3)**
>
> > **W2-3: Adding an OOD experiments and discuss generalization could be beneficial.**
>
> As the reviewer points out, performance in out-of-distribution (OOD) settings is still an important issue in offline RL. However, the main focus of this work was on addressing challenges of offline goal-conditioned RL, such as stitching, long-horizon reasoning, and multi-goal generalization. OGBench was also designed to target these aspects rather than setups for OOD environments. The limitation section explicitly states that the performance of CoGHP may degrade when moving to environments or tasks that are substantially different from the training data.
>
> However, CoGHP can handle OOD goals to some extent. In the OGBench Cube environment, the offline dataset is collected by a random pick-and-place policy, so specific evaluation goal configurations may be rarely, if ever, observed in the data. As described in the OGBench paper, this setting can therefore be viewed as requiring some degree of unseen goal generalization. In our experiments, CoGHP achieves consistently higher success rates than previous offline GCRL algorithms on several variants of Cube (cube-single, cube-double, cube-triple), which suggests that, at least within the same environment, CoGHP has a certain level of unseen goal generalization ability for goal configurations that appear only rarely in the data.
>
> > **W2-4 & Q1: The reason behind generating subgoal in the order of farthest to nearest is not very clear.**
>
> Thank you for your question regarding the ordering of subgoal generation. Our initial design assumed that subgoals closer to the current state should aggregate more comprehensive information from the hierarchical reasoning process, and we therefore generated subgoals from the one farthest from the current state to the one closest to it. Thus, the ordering we adopt is based on this assumption rather than reflecting any structural difficulty in the architecture. To test this assumption, we added an ablation that compares forward-order generation, which generates from the subgoal closest to the current state to the farthest subgoal, with reverse-order generation. We conducted experiments on navigation tasks where generating multiple subgoals yields stable performance, and examined how environment difficulty and the number of generated subgoals $H$ affect the impact of the subgoal generation order (`Appendix C.5`). In the easier antmaze-large environment, forward and reverse generation perform similarly. In contrast, in antmaze-giant, reverse generation outperforms forward generation, and the performance gap widens as $H$ increases. These findings support the validity of our initial design assumption.
>
> |  | forward (H=2) | forward (H=5) | reverse (H=2) | reverse (H=5) |
> | --- | --- | --- | --- | --- |
> | antmaze-large-navigate-v0: | 90$\pm$2 | 92$\pm$2 | 90$\pm$3 | 92$\pm$2 |
> | antmaze-giant-navigate-v0: | 71$\pm$2 | 50$\pm$5 | 78$\pm$8 | 61$\pm$7 |
>
> [1] Zawalski, Michał, et al. "Robotic control via embodied chain-of-thought reasoning." *arXiv preprint arXiv:2407.08693* (2024).
> [2] Zhao, Qingqing, et al. "Cot-vla: Visual chain-of-thought reasoning for vision-language-action models." *Proceedings of the Computer Vision and Pattern Recognition Conference*. 2025.
> [3] Chen, Lili, et al. "Decision transformer: Reinforcement learning via sequence modeling." Advances in neural information processing systems 34 (2021): 15084-15097.
> [4] Janner, Michael, Qiyang Li, and Sergey Levine. "Offline reinforcement learning as one big sequence modeling problem." Advances in neural information processing systems 34 (2021): 1273-1286.

---

### Author Response · Authors · 2025-11-21
**General Response**

We sincerely thank all the reviewers for their valuable feedback. Your comments have helped us address the weaknesses in our paper and significantly improve its overall quality. In this section, we provide a summary of the revisions made based on your suggestions.

> **Paper revision summary**

All revisions have been highlighted in blue text.

- (For reviewer yZJW, chFA, DvEG, 7rmc) We have added further explanation of the relationship between chain-of-thought and CoGHP in Section 1.
- (For reviewer 7rmc) We updated “stepwise procedural consistency” in Section 4.1 to the clearer term “position-dependent token roles” and explained this meaning explicitly.
- (For reviewer chFA, 7rmc) We have revised Section 4.4 and Appendix D to explicitly state that extending the framework to other subgoal types is left as future work.
- (For reviewer chFA, 7rmc) We have added the experimental setup and analysis of the Transformer baseline in Appendices B.3 and C.4.
- (For reviewer yZJW, chFA) We have added Appendix C.3 to report a sensitivity analysis of the joint variation of subgoal count and subgoal step.
- (For reviewer yZJW, DvEG, 7rmc) We have added an ablation on subgoal generation order in Appendix C.5.
- (For reviewer chFA) We have added an ablation on causal mixer variants in Appendix C.6.
- (For reviewer chFA) We have added a teacher forcing ablation in Appendix C.8.
- (For reviewer chFA) We have added an advantage-temperature ablation in Appendix C.9.
- (For reviewer yZJW, chFA, 7rmc) We have added visualizations of subgoals in the visual-antmaze environment in Appendix C.10.
- (For reviewer 7rmc) We have added a discussion of alternative formulations for subgoal generation in Appendix D.

---

### Author Response · Authors · 2025-12-04
**Final Remarks on CoGHP and Revisions**

We sincerely thank the reviewers and the AC for carefully reading our paper and providing thoughtful feedback. We are pleased that the reviewers have recognized CoGHP as a strong alternative to conventional two-level hierarchical architectures for long-horizon offline goal-conditioned RL. In particular, we appreciate the comments highlighting that our single MLP-Mixer–based policy, which autoregressively generates subgoals while preserving access to the final goal and enabling gradients to flow through the entire hierarchy, is both conceptually clean and practically useful, and that CoGHP achieves substantial performance gains over strong existing baselines across diverse domains.

**In this work, we made, to the best of our knowledge, the first attempt to introduce both chain-of-thought–style reasoning and the MLP-Mixer architecture into offline goal-conditioned RL. Our main goal in doing so was to mitigate the structural limitations of prior offline hierarchical RL methods that relied on separately trained two-level networks.** We consider “thinking step-by-step” to be one of the core aspects of chain-of-thought reasoning, and we instantiate this idea in CoGHP by (i) unfolding challenging goal-reaching problems into a sequence of latent subgoal tokens, (ii) keeping this sequence conditional on the final goal while generating primitive actions through these intermediate subgoals, and (iii) processing the whole hierarchy end-to-end within a single unified model. To realize this hierarchical policy in practice, CoGHP uses an MLP-Mixer backbone, which enables efficient cross-token communication, unified autoregressive generation, and end-to-end optimization. This chain-of-thought-style “input - intermediate reasoning steps - action” structure aligns with recent VLA-based chain-of-thought approaches. At the same time, CoGHP takes a step further by extending it to a hierarchical policy learning framework for offline goal-conditioned RL.

**Building on the reviewers’ insightful suggestions, we strengthened the revised draft through conceptual clarifications, additional ablations, and more detailed architectural analyses.** We clarified the conceptual connection between chain-of-thought and CoGHP, explicitly stating which aspects of chain-of-thought our architecture implements. Experimentally, we added (i) an ablation on subgoal generation order, providing empirical support for our assumption that reverse-order generation (from the farthest to the nearest subgoal) is more effective, (ii) ablations on the causal mixer, including variants without the mixer and with fixed lower-triangular averaging, (iii) a comparison between training with and without teacher forcing, and (iv) sensitivity analyses over the advantage temperature, subgoal loss weight, subgoal step k, and the number of subgoals H. We also provided a more detailed comparison between MLP-Mixer and Transformer backbones, which supports our claim that the MLP-Mixer architecture better matches the characteristics of CoGHP.

We believe that these clarifications and additional experiments have addressed the reviewers’ concerns and further highlighted the strengths of CoGHP as a new offline goal-conditioned RL approach for long-horizon problems. We would be grateful if you could take our responses to the reviewers’ feedback into account.

---

### Meta-Review · Area_Chair_m7Ym · 2026-01-12

**Summary:**

This paper introduces chain-of-thought–style reasoning and the MLP-Mixer architecture into offline goal-conditioned RL. The reviewers have concerns regarding (i) misleading use of the term chain-of-thought, (ii) incomplete ablation studies and limited experimental scope, and (iii) potential distribution shift vulnerability.

**Reviewer Concerns:**

Although the authors provide responses trying to address all the above concerns, it seems to me that the experiments are still limited to OGBench, and experiments on out-of-distribution generalization are still limited. Given the high bar of ICLR, I would suggest rejection and encourage the authors to add more experiments on the above two aspects.

**Reviewer Scores:**

It is unclear whether the reviewers would have changed their scores, however I would suspect the two reviewers with negative scores might not change their scores even with a full discussion phase.

---

### Decision · Program_Chairs · 2026-01-26

Reject